# HOI-Swap: Swapping Objects in Videos with Hand-Object Interaction Awareness

**Zihui Xue[1,2]**   **Mi Luo[1]**   **Changan Chen[1]**   **Kristen Grauman[1,2]**
[1]The University of Texas at Austin   [2]FAIR, Meta

## Abstract

We study the problem of precisely swapping objects in videos, with a focus on those interacted with by hands, given one user-provided reference object image. Despite the great advancements that diffusion models have made in video editing recently, these models often fall short in handling the intricacies of hand-object interactions (HOI), failing to produce realistic edits—especially when object swapping results in object shape or functionality changes. To bridge this gap, we present HOI-Swap, a novel diffusion-based video editing framework trained in a self-supervised manner. Designed in two stages, the first stage focuses on object swapping in a single frame with HOI awareness; the model learns to adjust the interaction patterns, such as the hand grasp, based on changes in the object's properties. The second stage extends the single-frame edit across the entire sequence; we achieve controllable motion alignment with the original video by: (1) warping a new sequence from the stage-I edited frame based on sampled motion points and (2) conditioning video generation on the warped sequence. Comprehensive qualitative and quantitative evaluations demonstrate that HOI-Swap significantly outperforms existing methods, delivering high-quality video edits with realistic HOIs.[1]

## 1 Introduction

Consider a video depicting the process of a human hand picking up a kettle and moving it around (Figure 1). What if we want to replace the hand-interacting object in this scene with another item specified by the user—perhaps a differently-shaped kettle, a bottle, or a bowl? This capability—to swap the *in-contact* object in a video with another, while aligning with the original video's content—is crucial for enhancing various real-world applications. Such functionality can transform entertainment, allowing users to create novel video content without the need to re-record or engage in labor-intensive manual editing. For example, in advertising, there may be situations where a pre-recorded video needs to adapt to new sponsorship requirements by replacing a soda can in the video with a water bottle. Additionally, it holds significant promise in robotics, where recent results suggest generative models can reduce the reliance on manually collected task-specific visual data and thereby enable large-scale robot learning [25, 58]. For example, imagine a scenario where, from just a single video of a mug being picked up, a generative model is able to produce numerous variants of this video with diverse objects such as bottles, bowls and kettles. This capability could greatly streamline the data collection process, reducing the need for extensive manual data collection.

However, hands are notoriously challenging to work with in image/video editing [62, 66]. They pose significant hurdles in manual photoshopping and often produce unsatisfactory outputs when automated by generative models. The precise swapping of hand-interacting objects presents a unique challenge that existing diffusion models [19, 45, 1, 56], despite their advances in video editing, fail to address adequately. This difficulty arises from three main factors: the need for (a) HOI-aware capabilities, (b) spatial alignment with the source context, and (c) controllable temporal alignment with the source video's motion patterns.

---

[1]Project webpage: https://vision.cs.utexas.edu/projects/HOI-Swap.

38th Conference on Neural Information Processing Systems (NeurIPS 2024).

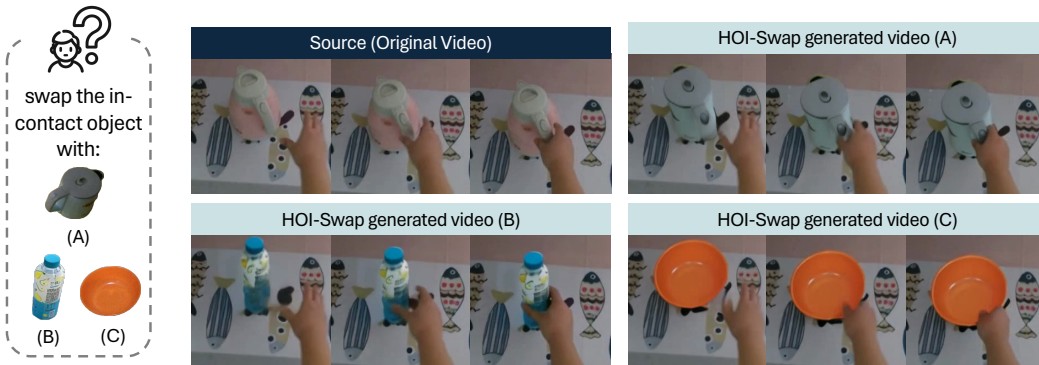

Figure 1: We present HOI-Swap that seamlessly swaps the *in-contact* object in videos using a reference object image, producing precise video edits with natural hand-object interactions (HOI). Notice how the generated hand needs to adjust to the shapes of the swapped-in objects (A,B,C) and how the reference object may require automatic re-posing to fit the video context (A).

First, consider the challenge of HOI awareness. Objects chosen for swapping often vary in their properties from the original, resulting in changes to interaction patterns. For example, as illustrated in Figure 2 (a), replacing the kettle from the original video with a bowl necessitates adjustments in the hand's grasp patterns. While many generative inpainting methods [57, 50, 36, 7, 51, 16] have been developed to insert reference objects into specific scene regions, they are generally limited to contexts where the objects are *isolated*—not in contact with a human hand or other objects—and thus lack HOI awareness. In Figure 2 (a), the two image inpainting approaches Paint by Example (PBE) [57] and AnyDoor [7] either merely replicate the hand pose from the original image, or produce unnaturally occluded hands, resulting in suboptimal and unrealistic HOI patterns.

Second, consider the challenge of spatial alignment with the original video. The reference object might appear in any arbitrary pose; for instance, in Figure 2 (b), the kettle handle is on the left in the reference image, but for realistic interaction, the generated content needs to reposition the kettle handle to the right, where the hand is poised to grasp it. However, current approaches do not offer this level of control, as evidenced by the results from a hand insertion approach Affordance Diffusion (Afford Diff) [62] in Figure 2 (b). Despite being adept at generating a hand interacting with the given kettle, it does not ensure correct object placement to align with the hand and scene context in the original image, lacking spatial alignment capability.

Third, consider the challenge of temporal alignment with the original video. We highlight a crucial observation in this problem: the motion information in an HOI video sequence is closely tied to the object's characteristics (such as its shape and function). This means that when swapping objects, *not all motions from the source video are appropriate or transferable* to the new object. For example, Figure 2 (c) shows a hand closing a trash can, alongside an HOI-Swap edited image where the original can is replaced with one that is differently shaped and functions differently. Ideally, the generated content should reflect the motion of closing the lid, yet it may not replicate the exact motion from the source video due to these differences. Conversely, Figure 1 (first row) depicts a scenario of swapping one kettle with another. Here, the objects undergo only slight shape changes, allowing the generated video to closely follow the source video's motion. While there are varying degrees to which object changes can affect the original motion, current video editing approaches [42, 12, 10, 59, 21, 6, 5, 34] adhere to a rigid degree of motion alignment, often targeting 100%, adopting conditional signals like optical flow or depth sequences that encode substantial object information. Consequently, they lack the controllability to adjust the degree of motion alignment based on object changes.

To address these challenges, we introduce HOI-Swap, a video editing framework designed for precise object edits with HOI awareness. We approach the challenge by posing it as a video inpainting task, and propose a fully self-supervised training framework. Our innovative approach structures the editing process into two stages. The first stage addresses HOI awareness and establishes spatial alignment, by training an image inpainting diffusion model for object swapping in one frame. The second stage propagates this one-frame edit across the remaining frames to achieve temporal alignment with the source. For this, we propose to warp a video sequence from the edited frame using randomly sampled points tracked across frames with optical flow, and train a video diffusion model that learns to fill in

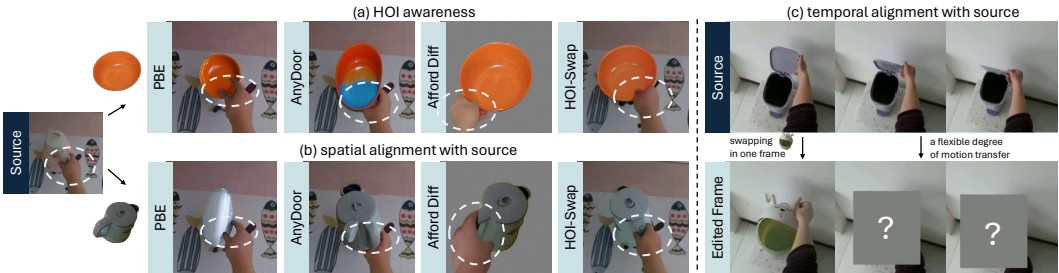

Figure 2: We highlight three challenges for the in-contact object swapping problem: (a) HOI awareness, where the model needs to adapt to interactions, such as changing the grasp to realistically accommodate the different shapes of the kettle vs. the bowl; (b) spatial alignment with source, requiring the model to automatically reorient objects, such as aligning the blue kettle from the reference image to match the hand position in the source; (c) temporal alignment with source, necessitating controllable motion guidance capability, essential when swapping objects like a trash can with a differently shaped and functioning reference, where not all original motions are transferable or desirable. In (a) and (b), we compare HOI-Swap's edited images with Paint by Example (PBE) [57], AnyDoor [7], and Affordance Diffusion (Afford Diff) [62].

the sparse and incomplete warped sequence. Our approach thus enables *controllable* motion alignment by varying the sparsity of sampled points. During inference, users can modulate this number based on the object's changes—sampling fewer or no points for significant shape or functional alterations, and more points for minor changes to closely replicate the source video's motion. HOI-Swap is evaluated on both image and video editing tasks, consistently producing high-quality edits with realistic HOIs. It greatly surpasses existing editing approaches in both qualitative and quantitative evaluations, including a user study. By extending the capabilities of generative models into the largely unexplored domain of HOI, we hope that HOI-Swap opens new avenues for research and practical applications in this innovative field.

## 2 Related Work

**Generative Models for Image Editing** Recent advances in diffusion models [19, 45] have significantly enhanced the capabilities of image editing. Predominant models [38, 18, 2, 26, 53, 24] use *text* as guidance, which, despite its utility, often lacks the precision needed for exact control. In response, a growing body of studies have begun to explore the use of reference images as editing guidance. Customized approaches like Textual Inversion [11] and DreamBooth [47] are designed to generate new images of a specific object given several of its images and a relevant text prompt. However, these methods require extensive finetuning for each case and lack the ability to integrate the object into another user-specified scene image. More closely related to our task, a few approaches aim to seamlessly blend a reference object [57, 50, 36, 7, 51, 16] or person [29] into a specific region of a target scene image. However, as demonstrated in Figure 2 and detailed in Section 4, these methods prove inadequate, often producing images with unnatural HOIs, such as missing fingers, distorted hands, or oddly shaped objects. These issues reveal shortcomings in current generative models, and motivate the development of HOI-Swap.

**Generative Models for Video Editing** With the advent in diffusion-based text-to-image and text-to-video generation [45, 1, 13], many efforts explore extending pretrained diffusion models for video editing, employing zero-shot [12, 59, 42, 4, 60, 28], or one-shot-tuned learning paradigms [55, 32, 17, 67]. However, these methods require extensive video decomposition or costly per-video fine-tuning, with processing times ranging from several minutes to multiple hours on high-end GPUs for a single edit, which curtails their usability in practical creative tools. Another line of work [10, 6, 34, 54, 63, 41] adopts a training-based approach, where models are trained on large-scale datasets to enable their use as immediately effective editing tools during inference. Our work falls into this training-based paradigm.

Similar to image editing, the majority of video editing approaches [1, 13, 12, 59, 42, 4, 60, 55, 32] rely on *text* as editing guidance. Since text prompts may not accurately capture the user's intent [67, 14, 41], for our task, using an object image provides more precise guidance. In terms of motion guidance,

most video editing approaches [42, 12, 10, 59, 21, 6, 5, 34, 63] utilize per-frame structural signals extracted from the source video, including depth map, optical flow, sketches, or canny edge sequences, facilitated by well-trained spatial-control image diffusion models such as ControlNet [65], T2I-Adapter [39] and Composer [22]. Since these structural signals inherently encode the shape of the original object, they are unsuitable for edits involving shape changes [41, 17]. Recent works explore shape-aware video editing, by the use of Layered Neural Atlas [32], modifying the source video's optical flow sequences [41], or establishing correspondence with sparse semantic points [17]. However, all these approaches overlook the impact of object changes on motion and only enforce a fixed degree of motion alignment. In contrast, HOI-Swap introduces flexibility by allowing users to adjust the sparsity of sampled points during inference, providing precise motion control.

**Generating Hand-Object Interactions** There is growing interest in generating plausible HOIs [20, 31, 62, 61, 9, 66, 64]. Affordance Diffusion [62] inserts a synthesized hand given a single object image. Similarly, HOIDiffusion [66] aims to create HOI images conditioned on a 3D object model along with detailed text descriptions. Meanwhile, other approaches in the 3D domain focus on generating realistic 3D HOI interactions from textual descriptions [9], synthesizing grasping motions [64] or reconstructing 3D HOIs from real videos [61]. Despite these advancements, the task of accurately swapping in-contact objects in videos remains unexplored. Our work directly addresses this gap.

# 3 Method

We first formulate the problem and provide an overview of the two-stage HOI-Swap in Section 3.1. Section 3.2 and Section 3.3 detail the first and second stage of HOI-Swap, respectively.

## 3.1 Task Formulation and Framework Overview

Given a source video $\mathcal{V} = \{\mathcal{I}_i\}_{i=1}^N$ consisting of $N$ frames (where $\mathcal{I}_i$ represents the $i$-th frame), a binary mask (bounding box) sequence $\{\mathcal{M}_i\}_{i=1}^N$ that identifies the region of the source object to be replaced in $\mathcal{V}$, and an image of the reference object $\mathcal{I}^{ref}$, the objective is to generate a modified video $\mathcal{V}^\star$, that seamlessly swaps the original object in $\mathcal{V}$ with $\mathcal{I}^{ref}$.

We create a fully self-supervised training approach, necessitated by the impracticality of collecting paired videos $(\mathcal{V}, \mathcal{V}^\star)$. We pose the problem as a video inpainting task. Specifically, from the original training video $\mathcal{V}$ and mask $\mathcal{M}$, we derive a masked video $\mathcal{V}^m$, accompanied by a set of reference object images $\{\mathcal{I}^{ref}\}$. During training, the model takes the masked video sequence $\mathcal{V}^m$ and an image $\mathcal{I}^{ref}$ of the same object originating from a different random timepoint (and hence varying pose and/or viewpoint) in the same training video to reconstruct $\mathcal{V}$. During inference, the model is given a bounding box-masked sequence combined with various object images to test its swapping capability.

As outlined in Section 1, the task presents three main challenges: HOI awareness, spatial and temporal alignment. In response, we propose a two-stage framework that decomposes the complexity. The first stage integrates the reference object into a single frame, targeting HOI awareness and establishing spatial alignment initially. The second stage propagates the one-frame edit across the entire video sequence, focusing on temporal alignment and carrying forward the spatial alignment from the first stage. This structured approach effectively lightens the model's generation load. In terms of which frame to select from the input video for stage-I edits, we automate the process by adopting an off-the-shelf hand-object detector [49] to identify frames with hand-object contact, from which we randomly select one for editing. See Supp. C.4 for a detailed analysis. For greater applicability and flexibility, each stage is trained separately in a self-supervised manner, using the original object image as $\mathcal{I}^{ref}$. Consequently, the stage-I model not only plays a crucial role in the video editing pipeline but also serves as a standalone image editing tool. The full pipeline is illustrated in Figure 3.

## 3.2 Stage I: HOI-aware Object Swapping in One Frame

In the first stage, given a source frame $\mathcal{I}_i$, a binary mask $M_i$ (where 1 indicates the object's region and 0 denotes the background) and a reference object image $\mathcal{I}^{ref}$, the objective is to generate an edited frame $\mathcal{I}_i^\star$ where the reference object is naturally blended into the source frame. We aim for the generated content $\mathcal{I}_i^\star$ to exhibit HOI awareness and spatial alignment with $\mathcal{I}_i$, ensuring that the

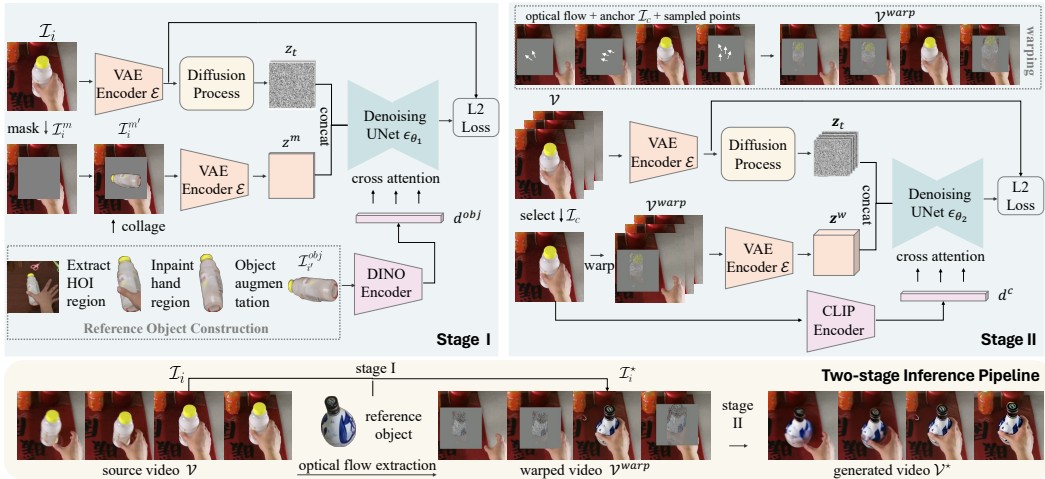

Figure 3: HOI-Swap involves two stages, each trained separately in a self-supervised manner. In stage I, an image diffusion model $\epsilon_{\theta_1}$ is trained to inpaint the masked object region with a strongly augmented version of the original object image. In stage II, one frame is selected from the video to serve as the anchor. The remaining video is then warped using this anchor frame, several points sampled within it, and optical flow extracted from the video. A video diffusion model $\epsilon_{\theta_2}$ is trained to reconstruct the full video sequence from the warped sequence. During inference, the stage-I model swaps the object in one frame. This edited frame then serves as the anchor for warping a new video sequence, which is subsequently taken as input for the stage-II model to generate the complete video.

reference object is realistically interacting with human hands and accurately positioned within the scene context of the source frame.

**Constructing paired training data**  Training is conducted in a self-supervised manner. Below, we detail how we prepare pseudo data pairs for training.

- **Masked frame preparation**: For each frame $\mathcal{I}_i$, we obtain the object's bounding box (denoted by $\mathcal{M}_i^{bbox}$) from its segmentation mask $\mathcal{M}_i$. To prevent the bounding box shape from influencing the aspect ratio of the generated results, we adjust the bounding box to be square. The masked frame is thus derived by applying the square bounding box mask to the frame, yielding $\mathcal{I}_i^m = \mathcal{I}_i \odot M_i^{bbox}$. We then crop and center the original frame to focus on the object region, enhancing its visibility.

- **Reference object preparation**: We extract the object image $\mathcal{I}_i^{obj} = \mathcal{I}_i \odot (1 - \mathcal{M}_i)$, which may be incomplete or partially obscured due to contact with hands or other objects. To address these occlusions, we employ an off-the-shelf text-guided inpainting model [40] to fill in the missing parts of the object, using the object's name as the text prompt.

- **Object augmentation**: Directly forming a training pair $(\mathcal{I}_i^{obj}, \mathcal{I}_i^m \rightarrow \mathcal{I}_i)$ is suboptimal as it does not reflect the variability expected in real-world applications. At test time, the reference object may appear in any pose, orientation, or size. To bridge this gap between training and testing scenarios, we apply strong augmentation techniques: (1) Spatially, we enhance the object's diversity by applying random resizing, flips, rotations, and perspective transformations; (2) Temporally, instead of using the object image from its original frame $i$, we introduce variability by randomly selecting an alternate frame, $\mathcal{I}_{i'}^{obj}$, from all available frames in the source video that contain the object. Finally, for preservation of the reference object's identity and structural details, we collage the augmented reference object image $\mathcal{I}_{i'}^{obj}$ onto the masked region in $\mathcal{I}_i^m$, producing $\mathcal{I}_i^{m'}$.

**Model design**  We design the stage-I model as an image-conditioned latent diffusion model (LDM). Specifically, we employ a pretrained variational autoencoder (VAE) [27] $\mathcal{E}$ as used in prior work [45] to encode a frame $\mathcal{I}$ into a latent space representation. This encoding is denoted by $z = \mathcal{E}(\mathcal{I})$, where $\mathcal{I} \in \mathbb{R}^{3 \times H \times W}$, $z \in \mathbb{R}^{4 \times H/8 \times W/8}$, $H$ and $W$ denote the frame's height and width, respectively. During the forward diffusion process, Gaussian noise is gradually added to $z$ over $T$ steps, producing a sequence of noisy samples $\{z_0, \cdots, z_t, \cdots, z_T\}$. We train a denoising network $\epsilon_{\theta_1}(\circ, t)$ that learns to reverse the diffusion process, which is implemented as a UNet [46]. The training objective of stage

I is summarized as:

$$\mathcal{L}_{stageI} = \mathbb{E}_{z, z^m, d^{obj}, \epsilon \sim \mathcal{N}(0,1), t} \left[ \| \epsilon - \epsilon_{\theta_1}(z_t, z^m, d^{obj}, t) \|_2^2 \right]. \qquad (1)$$

Our LDM is designed to take in two types of conditional signals alongside the standard $z_t$ and $t$: (1) The reference object image $\mathcal{I}_{i'}^{obj} \in \mathbb{R}^{3 \times H \times W}$ is encoded through DINO [3] for distinctive object features $d^{obj} \in \mathbb{R}^{768}$, $d^{obj}$ is then taken by $\epsilon_{\theta_1}$ as input to guide the denoising process via cross-attention mechanisms; (2) The masked frame $\mathcal{I}^m \in \mathbb{R}^{3 \times H \times W}$ is encoded by the same VAE to produce $z^m = \mathcal{E}(\mathcal{I}^m)$. $z^m \in \mathbb{R}^{4 \times H/8 \times W/8}$ is then concatenated channel-wise with $z$ before being fed into $\epsilon_{\theta_1}$.

### 3.3 Stage II: Controllable Motion-guided Video Generation

In stage II, given the first-stage edit $\mathcal{I}_i^\star$, source video $\mathcal{V}$, and its binary mask sequence $\mathcal{M}$, the objective is to generate a new video $\mathcal{V}^\star$ that propagates the single-frame edit across the remaining frames. For this purpose, we perform warping to transfer pixel points from the edited frame to the remaining frames, resulting in a sparse and incomplete video sequence. We then train a video diffusion network that learns to correct and fill in the gaps, completing the video editing process.

**Constructing paired training data** Training continues in a self-supervised manner. For this purpose, we use the original frame $\mathcal{I}_i$ from $\mathcal{V}$ during training and only replace $\mathcal{I}_i$ with the stage-I edit $\mathcal{I}_i^\star$ during inference time. The detailed process is explained below:

- **Masked frame sequence preparation**. We employ the same frame masking strategy in stage I and apply it to a sequence of frames $\mathcal{V} = \{\mathcal{I}_i\}_{i=1}^N$. To standardize the masking across the sequence, we identify the largest bounding box $\mathcal{M}_{max}^{bbox}$ from the sequence $\{\mathcal{M}_i^{bbox}\}_{i=1}^N$ and mask each frame $i$ by $\mathcal{I}_i^m = \mathcal{I}_i \odot \mathcal{M}_{max}^{bbox}$, resulting in a masked frame sequence $\{\mathcal{I}_i^m\}_{i=1}^N$. This ensures that the model is trained to inpaint a consistent object region across different frames.
- **Conditioning frame selection**. To avoid limiting the model to generating videos based on a specific reference frame, such as the first frame as often used in existing image-to-video generative models, we randomly select a frame $\mathcal{I}_c$ from $\mathcal{V}$ as the conditioning frame. This any-frame-conditioning mechanism brings additional flexibility during inference (for detailed discussion, see Supp. C.4).

**Controllable motion guidance** Given that object swapping can result in changes to shape or functionality, our approach is designed to be flexible and adaptable, allowing for varying degrees of motion pattern encoding from the source video. The key idea is to control the level of motion agreement with the source via points sampled within the masked region. Tracking a large number of points over time captures extensive motion information from the source video, but it also reveals much about the source object's characteristics (e.g. shape). Conversely, using few points reduces the motion information and object characteristics carried over, but offers the model more freedom to generate plausible motion for the target. The latter scenario is particularly useful when only partial motion transfer is desired due to differences between objects, as exemplified in Figure 2 (c).

To be specific, our approach involves the following steps: (1) uniformly sample $r\%$ pixel points within the original object's region in the conditioning frame $\mathcal{I}_c$ (i.e., where $\mathcal{M}_{max}^{bbox}$ equals 1), (2) track these points using optical flow vectors computed by RAFT [52], and (3) warp a new video sequence based on these tracked points, using $\mathcal{I}_c$ as the anchor. The resulting warped sequence is denoted by $\mathcal{V}^{warp} = \{\mathcal{I}_i^{warp}\}_{i=1}^N$. The anchor frame $\mathcal{I}_c^{warp}$ is intact and serves as the basis for warping (i.e., $\mathcal{I}_c^{warp} = \mathcal{I}_c$). For the other frames ($i \neq c$), we overlay these points tracked from the conditioning frame $c$ to frame $i$ onto the masked region of $\mathcal{I}_i$ (i.e., $\mathcal{I}_i^m$), creating a composite image $\mathcal{I}_i^{warp}$. See Figure 3 (upper right) for an illustration of this process.

We vary the sparsity level of sampled motion points, $r$ from 0 to 100 during training, preparing the model for a range of scenarios—from no motion guidance to full motion information. During inference, we demonstrate that the model is capable of generating video sequences with varying levels of motion alignment to the original video (see Section 4.2 and Supp. C.4 for further discussion). Thus, our stage-II training goal is to take the warped yet incomplete video sequence ($\mathcal{V}^{warp}$) as input, then learn to fill in the gaps and smooth over discontinuities to reconstruct the full video ($\mathcal{V}$).

**Model design** We design the stage-II model as a video LDM. The LDM architecture and training objective closely follow those of stage I, with some adaptations for handling video sequences. We continue to use the VAE encoder $\mathcal{E}$, but now apply it to a sequence of frames $\mathcal{V} \in \mathbb{R}^{N \times 3 \times H \times W}$

to obtain the latent feature $\mathbf{z} \in \mathbb{R}^{N \times 4 \times H/8 \times W/8}$. We inflate the 2D UNet into a 3D UNet as the denoising network (denoted by $\epsilon_{\theta_2}(\circ, t)$) for generating a frame sequence. Specifically, we insert additional temporal layers into the 2D UNet and interleave them with spatial layers, following [1, 10]. The stage-II training objective is shown below:

$$\mathcal{L}_{stageII} = \mathbb{E}_{\mathbf{z}, \mathbf{z}^w, d^c, \epsilon \sim \mathcal{N}(0,1), t} \left[ \|\epsilon - \epsilon_{\theta_2}(\mathbf{z}_t, \mathbf{z}^w, d^c, t)\|_2^2 \right]. \tag{2}$$

Our video LDM involves two conditional signals in addition to the standard $\mathbf{z}_t$ (a noised version of $\mathbf{z}$) and diffusion time step $t$: (1) The warped video sequence $\mathcal{V}^{warp} \in \mathbb{R}^{N \times 3 \times H \times W}$, is processed similarly as $\mathcal{V}$, encoded by $\mathcal{E}$, producing a latent feature $\mathbf{z}^w \in \mathbb{R}^{N \times 4 \times H/8 \times W/8}$. $\mathbf{z}^w$ is then concatenated channel-wise with $\mathbf{z}$ before being fed into the denoising network $\epsilon_{\theta_2}$. (2) The selected frame $\mathcal{I}_c$, besides serving as the anchor in the warping process, is also encoded by CLIP [43], yielding $d^c \in \mathbb{R}^{768}$ that provides high-level contextual guidance and is incorporated in $\epsilon_{\theta_2}$ via cross-attention. Additionally, please see Supp. C.4 for an ablation of comparing CLIP and DINO as the object encoder.

## 4 Experiments

To comprehensively evaluate HOI-Swap, we consider both the image editing (accomplished by the stage-I model) and video editing (facilitated by the entire pipeline) task. We describe the experimental setup in Section 4.1 and present editing results along with an ablation analysis in Section 4.2.

### 4.1 Experimental Setup

**Datasets** Our training leverages two large-scale egocentric datasets, HOI4D [35] and Ego-Exo4D [15], which feature abundant HOIs, making them particularly suitable for exploring this problem. In total, the stage-I model is trained on 106.7K frames and the stage-II model uses 26.8K 2-second video clips for training. The evaluation set for image editing includes 1,250 source images, each paired with four reference object images, resulting in a total of 5,000 edited images. The evaluation set for video editing is composed of 25 source videos, also combined with four object images each, yielding 100 unique edited videos. Note that: (1) For image editing evaluation, we include both hand-present scenarios, which challenge the model's HOI-aware capabilities, and non-hand-present scenarios (20%) to evaluate its general object-swapping capability. (2) For video editing evaluation, in addition to HOI4D videos, we include challenging in-the-wild videos from EPIC-Kitchens [8] and third-person videos from TCN Pouring [48]—datasets not used in training—to assess the model's zero-shot generalization across diverse scenarios. See Supp. B.1 for complete dataset descriptions.

**Baselines** For image editing, our stage-I model is compared with two strong image inpainting approaches, PBE [57] and AnyDoor [7], along with a hand-insertion approach Afford Diff [62] in the HOI domain. For video editing, we consider the following baselines: (1) applying the best image editing baseline to each frame of the video; (2) integrating the best image editing baseline with AnyV2V [28], a recent video editing approach that takes one edited frame from any black-box image editing model as conditional guidance; (3) VideoSwap [17], the state-of-the-art approach in customized object swapping. Besides these object-image-guided approaches, we include a focused comparison with text-guided approaches for a thorough evaluation. See Supp. C.6 for the discussion.

**Evaluation** HOI-Swap is evaluated through both human and automatic metrics. We conduct a user study, involving 15 participants that assess 260 image edits and 100 video edits. Participants

Table 1: Quantitative evaluation of HOI-Swap using (1) automatic metrics: contact agreement (cont. agr.), hand agreement (hand agr.), hand fidelity (hand fid.), subject consistency (subj. cons.), and motion smoothness (mot. smth.)—the last two are for video only and (2) user preference rate (user pref.) from our user study. For video editing, users were given an option to report if they found all edits unsatisfactory, which has an average selection rate of 10.9%. All values are percentages.

| Image Editing | cont. agr. | hand agr. | hand fid. | user pref. | Video Editing | subj. cons. | mot. smth. | cont. agr. | hand agr. | hand fid. | user pref. |
|---|---|---|---|---|---|---|---|---|---|---|---|
| PBE [57] | 67.8 | 71.9 | 73.9 | 4.5 | Per-frame | 87.4 | 95.3 | 86.4 | 64.8 | 82.2 | 0.2 |
| AnyDoor [7] | 84.6 | 70.9 | 86.3 | 15.6 | AnyV2V [28] | 71.7 | 95.6 | 68.3 | 30.8 | 27.0 | 1.2 |
| Afford Diff [62] | 78.6 | 19.5 | 92.9 | 7.6 | VideoSwap [17] | 91.8 | 97.8 | 87.9 | 52.6 | 81.9 | 1.2 |
| HOI-Swap (ours) | **87.9** | **79.8** | **97.4** | **72.1** | HOI-Swap (ours) | **92.4** | **98.2** | **93.1** | **78.6** | **97.6** | **86.4** |

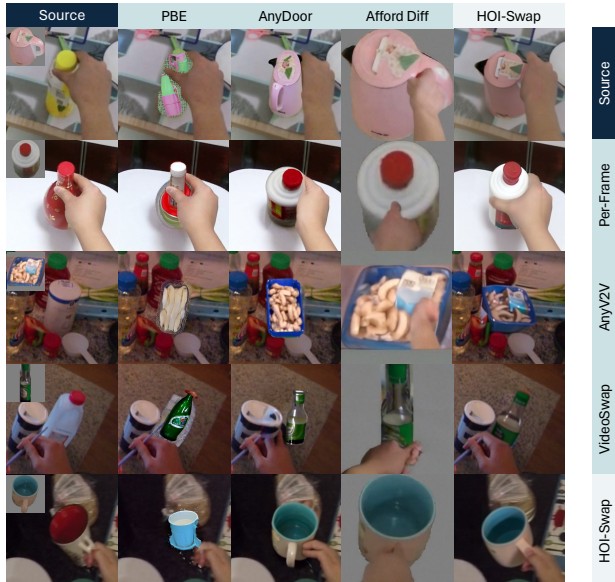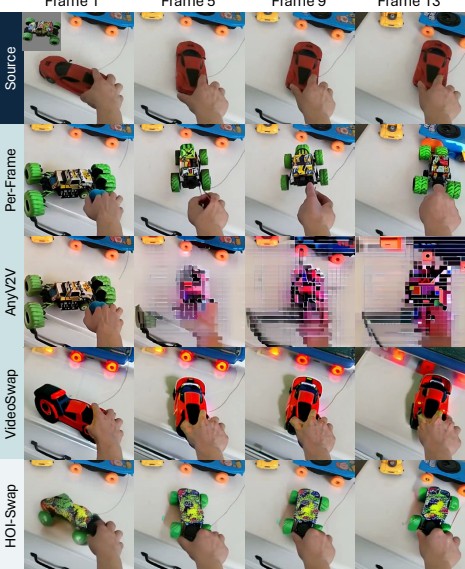

Figure 4: Qualitative results of HOI-Swap. We compare HOI-Swap with image (left) and video editing approaches (right). The reference object image is shown in the upper left corner of the source image. For image editing, HOI-Swap demonstrates the ability to seamlessly swap in-contact objects with HOI awareness, even in cluttered scenes. For video editing, HOI-Swap effectively propagates the one-frame edit across the entire video sequence while accurately following the source video's motion, achieving the highest overall quality among all methods. We highly encourage readers to check Supp. C.1 and the project page video for more comparisons.

are presented with four edited results, three from baselines and one from HOI-Swap (randomly shuffled) and asked to pick the best one in terms of object identity, HOI realism, and overall quality (for image edits), motion alignment and overall quality (for video edits). Each sample is assessed by three different participants. For automatic evaluation, we employ several metrics across different dimensions: (1) HOI contact agreement and hand agreement to measure spatial alignment, following [62, 66]; (2) hand fidelity, following [37]; (3) subject consistency and motion smoothness from VBench [23] to evaluate general video quality. See Supp. B.3 for full descriptions.

**Implementation** The stage-I model is trained on 512×512 resolution images for 25K steps. For stage-II training, input video resolution is set as 14×256×256, where we sample 14 frames at an fps of 7 and train the model for 50K steps. See Supp. B.4 for full details.

### 4.2 Editing Results

**Quantitative evaluation** Table 1 compares the performance of HOI-Swap on both image and video editing tasks. For image editing, the two inpainting approaches, PBE [57] and AnyDoor [7] struggle to generate the hand correctly, as indicated by hand fidelity scores below 90%. Afford Diff [62] is developed for the HOI context and also benefits from HOI-specific training data in HOI4D. It produces high-fidelity hands but does not address the editing task adequately, resulting in low contact and hand agreement with the source image. Our proposed HOI-Swap excels in both aspects, achieving superior performance in agreement and fidelity scores. In the human evaluation, HOI-Swap is consistently favored over the other three approaches, achieving a preference rate of 72.1%. For video editing, HOI-Swap outperforms all competing methods across all metrics, including general video quality metrics (subject consistency and motion smoothness) and HOI edit metrics (contact agreement, hand agreement and hand fidelity). It significantly surpasses the three baseline approaches, winning the user study with a selection rate of 86.4%. Recognizing the inherent challenges of the video editing task, we included a survey question asking users if "All edits are of very low quality; none successfully swap the object." This response was selected at an average rate of 10.9%. See Supp. C.3 for details.

**Qualitative evaluation** Figure 4 shows HOI-Swap's edited images (left) and videos (right) alongside baseline comparisons. For image editing, HOI-Swap exhibits strong HOI-awareness, accurately

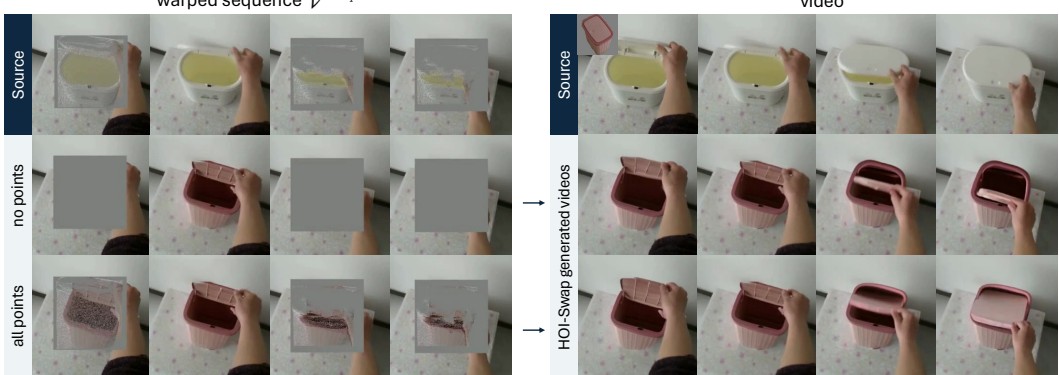

Figure 5: Ablation study on sampled motion points, comparing no to full motion points sampling. Left: we visualize $\mathcal{V}^{warp}$, used as conditional guidance for the stage-II model. Note that row 1 displays $\mathcal{V}^{warp}$ based on the source frame and is for illustration only, not provided to the model. Right: HOI-Swap exhibits controllable motion alignment: with no sampled points, the generated video diverges from the source video's motion; with full motion points, it closely mimics the source.

adjusting hand-holding patterns based on the reference object's properties (rows 1 and 2, with source images from HOI4D [35]). Moreover, HOI-Swap serves as a general object-swapping tool (not limited to HOIs) and can seamlessly swap in-contact objects in cluttered scenes (rows 3 and 4, with source images from EgoExo4D [15]). The final row demonstrates HOI-Swap's zero-shot generalization capability, using source images from EPIC-Kitchens [8]. Despite the out-of-distribution samples, HOI-Swap successfully aligns the reference mug handle with the source's hand context, whereas baselines like AnyDoor [7] fail to establish this connection. See Supp. C.1 for more examples.

For video editing (Figure 4 right), the source video depicts a hand rotating a toy car. The per-frame approach struggles with temporal consistency of the object identity and produces unnatural HOI motions. AnyV2V [28] fails to propagate the first-frame edit through subsequent frames, leading to inconsistencies. VideoSwap [17] preserves the overall look of the source video but fails to swap the reference object (i.e., the green toy car). In contrast, HOI-Swap delivers video edits that realistically depict the HOI and faithfully follow the source video's motion. Beyond this specific example, we note that baseline methods generally perform poorly on our in-contact object swapping task across various examples (see the project page video for more results). Despite our best efforts to reproduce their results using the official code and adjusting hyperparameters according to the provided guidelines, the performance of these models remained notably suboptimal. We observe similar issues with state-of-the-art text-guided approaches (Supp. C.6). This persistent underperformance highlights a big gap in current (otherwise quite powerful) diffusion models in their ability to accurately model the intricacies of HOIs, indicating great opportunities for further advancements in this field.

**Ablation study on motion points sparsity** HOI-Swap can generate videos with different degrees of motion alignment with the original video. Figure 5 compares the generated results when sampling no points versus full points. Here the source video depicts a closing trash can lid action. In row 2, when no points are sampled (resulting in no warping), the generated video does not follow the closing lid motion but instead depicts an action of putting the lid into the bin; the visible hand motion outside the inpainted region provides context, guiding the model to generate a plausible sequence with the hand moving downward. Conversely, using all motion points produces videos that closely align with the source video's motion. See Supp. C.4 for follow-up discussion on this.

**One-stage vs. two-stage design** Next we verify the efficacy of our two-stage design, where we train a one-stage model variant that takes the reference object image and the masked frame sequence as input and outputs the edited video. Figure 6 shows both qualitative and quantitative comparisons. The one-stage approach does not yield satisfactory results, specifically failing to preserve the reference object's identity. This limitation arises from the challenge of simultaneously addressing both spatial and temporal alignment, thereby reinforcing the advantage of our two-stage design.

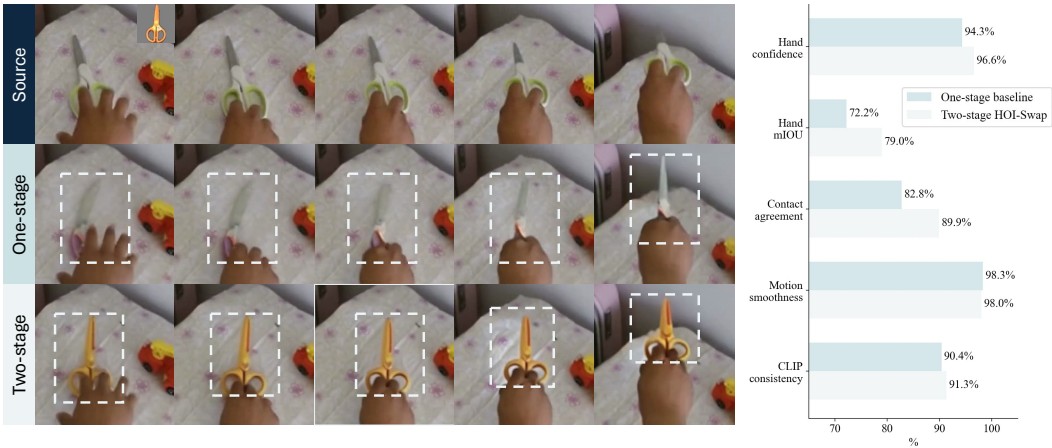

Figure 6: Qualitative and quantitative comparisons between a one-stage baseline [1] and our two-stage HOI-Swap. The one-stage model struggles with preserving the new object's identify and fails to generate accurate interaction patterns, yielding inferior quantitative performance.

Moreover, we present a comprehensive discussion on various design choices of HOI-Swap in Supp. C.4 and C.5, covering aspects such as object encoder choice, object masking strategy, editing frame selection strategy, and discussion on sampling regions.

## 5    Limitations and Future work

The problem of swapping *in-contact* objects in videos poses significant challenges. While HOI-Swap marks an important first step towards addressing these complexities, we recognize its current limitations. Specifically, we identify three areas for improvement: (1) **generalization to new objects.** HOI-Swap capably swaps new object instances with HOI awareness (e.g., swap an unseen mug with a differently-shaped bowl). More challenging scenarios involve presenting very different unseen reference object images for swapping. For example, the model needs to accurately depict a hand holding scissors, even though scissors were never part of the training dataset. This scenario requires the model to equip "world knowledge" that extends beyond its training data, enabling it to understand and realistically model how hands interact with a broader variety of objects. (2) **generalization to long video sequences with complex HOI.** The two-stage pipeline of HOI-Swap is motivated by the observation that HOI interactions remain stable throughout a short video clip. For instance, for a picking up mug sequence, replacing the mug with a bowl can be reliably done in one single frame and propagated across the remaining frames. However, for longer sequences, it is possible that an object undergoes multiple distinct hand interactions that change dynamically over time. This complexity necessitates the development of methods that can capture and model these varied HOI interactions across time. (3) **controllability**. Our proposed controllable motion guidance allows HOI-Swap to freely choose the degree of motion alignment with the source video in editing. One future work direction is to enhance this controllability with spatial support, allowing the model to specify which regions of the source video should be targeted for motion transfer. See Supp. D for more details on HOI-Swap's limitations and failure modes, including visual examples.

## 6    Conclusion

Recognizing the limitations of current diffusion models in effectively capturing HOIs, we introduce HOI-Swap, a novel approach for swapping hand-interacting objects in videos based on one reference object image. HOI-Swap consists of two stages: the first seamlessly integrates the reference object into a single frame, while the second propagates this edit across the entire sequence, with controllable motion alignment. Both qualitative and quantitative evaluations demonstrate that HOI-Swap produces realistic video edits with accurate HOIs that align well with the source content. In all, this work broadens the capabilities of generative models in video editing and represents the initial step towards solutions that can adeptly handle the intricacies of complex and dynamic HOIs in videos.

**Acknowledgements:** UT Austin is supported in part by the IFML NSF AI Institute. KG is paid as a research scientist at Meta. The authors would like to thank Zhengqi Gao (MIT), Xixi Hu (UT Austin), Hanwen Jiang (UT Austin), Feng Liang (UT Austin), and Yue Zhao (UT Austin) for their insightful discussions and valuable feedback, which contributed to the development of this work.

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

# Contents

## A    Supplementary Video

We invite readers to view the Supp. video available at `https://vision.cs.utexas.edu/projects/HOI-Swap`, which presents additional qualitative results of HOI-Swap. The video showcases video editing comparisons with baselines approaches, demonstrates zero-shot generalization on EPIC-Kitchens and TCN Pouring datasets (on which the model was not trained), includes an ablation study on varying the number of sampled motion points, and presents failure cases of HOI-Swap.

## B    Experimental Setup

### B.1    Datasets

We adopt all available videos in HOI4D [35] due to its comprehensive per-frame object mask annotations. We use 2,679 videos for training and hold out 292 videos for evaluation; the evaluation videos are selected based on object instance ids to ensure that the source objects during test time are unseen by the model. In total, the dataset offers 79.9K frames (sampled at the original 15 fps) and 26.8K 2-second video clips for training. HOI4D features 16 object categories, ranging from rigid objects like bowls, mugs, and kettles to articulated objects like laptops, pliers, and buckets. For evaluation, we select 1,000 images and 17 videos from the hold-out set as the source and combine each source with four randomly chosen reference object images.

For EgoExo4D [15], we utilize the ground truth object segmentation masks available at 1 fps for stage-I training. Since our video model operates at 7 fps, a future enhancement could involve interpolating these masks to scale up stage-II training with EgoExo4D videos and the corresponding pseudo masks. We focus on 18 frequently occurring object categories where editing is relevant: bowl, container, bottle, package, knife, pan, spoon, plate, spatula, pot, mug, cup, jar, glass, jug, kettle, and skillet. We adopt frames belonging to these categories and follow the official split, resulting in 18.7K training frames. For evaluation, we take 250 images from the official validation split as sources and combine each with four randomly chosen reference object images, resulting in 1,000 edited images.

Figure 7: Human evaluation interface for image editing part. We provide a source frame for editing alongside an image of the reference object. Users are asked to evaluate and select their favorite edited results based on various image editing criteria.

Additionally, we test the model's zero-shot editing capabilities on four EPIC-Kitchen [8] videos, which depict in-the-wild kitchen scenarios with often cluttered backgrounds, and four videos from TCN Pouring [48], with two being egocentric and two exocentric, featuring pouring motions. This evaluation encompasses a variety of challenging scenarios to thoroughly assess model performance.

## B.2 Baselines

We implement the baselines described in Section 4.1 using their official code. Baselines including PBE, AnyDoor and AnyV2V are adopted in a zero-shot manner. Afford Diff utilizes the same training data source (HOI4D) as ours. We do not provide additional layout or hand orientation information to Afford Diff. VideoSwap [17], a one-shot-tuned video editing approach, is trained on the source video using the default setup in their official repository. AnyDoor is recognized as the best image editing method and is integrated into the two video editing baselines, per-frame and AnyV2V.

Note that these baseline methods impose more demanding test-time requirements compared to ours. PBE and AnyDoor both necessitate precise segmentation masks of the source object, while HOI-Swap only requires a bounding box during inference. In addition, VideoSwap requires users to click semantic points for every video, provide a text prompt describing the video, and train a specialized object model using a few reference images. This extensive prepossessing limits its scalability and is a contributing factor to the smaller size of our video evaluation set. In contrast, HOI-Swap is fast on inference, with RAFT optical flow extraction completing in just a few seconds per video, making it efficient and practical for real-world applications.

## B.3 Evaluation

**Automatic evaluation**    For automatic evaluation, we employ several metrics across different dimensions: (1) To measure how well the generated content aligns spatially with the source image, we use **HOI contact agreement** and **hand agreement**. Following [62, 66], we utilize the in-contact state label provided by an off-the-shelf hand detector [49]. The HOI contact agreement score is determined

Figure 8: Human evaluation interface for video editing part. We provide a source video for editing alongside an image of the reference object. Users are asked to evaluate and select their favorite edited videos based on various video editing criteria.

by checking if the in-contact label of the generated frame matches the source frame. Additionally, we calculate mIOU between the generated and source hand regions using the detector's bounding boxes for hand agreement. (2) To evaluate how realistic the generated hand is, we follow [37] and assess **hand fidelity** based on the hand detector's confidence score. (3) To evaluate video quality, we consider **subject consistency** (DINO feature similarity across frames) and **motion smoothness** (determined by motion priors from a video frame interpolation model [33]) as proposed in VBench [23]. Note that we exclude object agreement from the automatic evaluation due to its inaccuracies. Often, the generated hand occludes some object regions, leading to lower similarity scores compared to simply placing the reference object in an area with no or partial hand presence. This discrepancy does not accurately reflect HOI realism and generation quality. To ensure a comprehensive evaluation, we incorporate questions in our user study to address this dimension.

**Human evaluation** We conducted a user study with 15 participants who assessed 260 image edits and 100 video edits randomly selected from the evaluation set. The human evaluation survey interfaces for image and video editing are shown in Figure 7 and Figure 8, respectively. Each survey began with general guidelines on the evaluation process, followed by examples of high-quality image or video editing results and good motion alignment for videos. For each question, participants were presented with four edited results: three from baselines and one from HOI-Swap, randomly shuffled. Each image or video sample was evaluated by three different participants to minimize bias.

For each image editing question, participants were asked to select the best one or two edits based on the following three evaluation dimensions:

1. **Object Identity:** The ability to preserve the identity of the reference object.

2. **HOI Realism:** The realism of the hand-object interaction and how well the edited image aligns with the reference object's semantics (e.g., hand grasping varies for a mug and a toy car) and the original frame's scene and hand context.

3. **Overall Quality:** The seamlessness of the object swap, considering:

- Does the generated object maintain the identity of the reference object image?
- How realistic is the hand-object interaction?
- How well does the edited image align with the reference object's semantics and the original frame's scene and hand context?

For each video editing question, participants were asked to select the best one or two edits based on the following two evaluation dimensions:

1. **Motion Alignment:** The extent to which the edit matches the motion of the source video.

2. **Overall Quality:** The seamlessness of the object swap, considering:
   - Does the hand-object interaction look realistic?
   - How well does the edited video align with the reference object's semantics and the original video's scene and hand context?
   - How well does the motion in the generated video align with the original video?

Recognizing the inherent challenges of the video editing task, we included a sanity check sub-question asking users if "All edits are of very low quality; none successfully swap the object."

### B.4 Implementation

For data preprocessing, we apply several transformations to the reference object image: random rotations with angles ranging from -90 to 90 degrees, random perspective changes, and random horizontal flips. We crop and center the source frame using a variable ratio between 0.3 and 0.6, determined during training to introduce variability; this ratio is defined as the size of the bounding box relative to the image size. For stage-I training: image resolution is set as $512 \times 512$. We train the model for a total of 25K steps with a learning rate of 1e-4 and a batch size of 8. We finetune the entire 2D UNet for image editing. For stage-II training: input video resolution is set as $14 \times 256 \times 256$, where we sample 14 frames at an fps of 7. The model is trained for a total of 50K steps with a learning rate of 1e-5 and a batch size of 1. We finetune the temporal layers of the 3D UNet. A classifier-free guidance dropout rate of 0.2 is employed for all stages. Training for each stage takes about 3 days on one 8-NVIDIA-V100-32G GPU node. We set sampling points sparsity value as 50% for all quantitative evaluations.

## C  Editing Results

### C.1  More qualitative results

Beyond the examples shown in Figure 4 of the main paper, we provide additional qualitative image editing results of HOI-Swap in Figures 9 and 10. HOI-Swap excels in both structured scenarios with relatively clean backgrounds (Figure 9, source frames from HOI4D and TCN Pouring), and challenging in-the-wild scenarios featuring cluttered backgrounds (Figure 10, source frames from EgoExo4D and EPIC Kitchens). Compared to the baselines, HOI-Swap seamlessly integrates the reference object into the target scene, demonstrating exceptional HOI awareness and the ability to accurately reorient the reference object to align with the scene context. Moreover, it also performs well in general object-swapping settings where no hand is involved, as shown in rows 3, 5, and 6 of Figure 10. Please refer to our project page for more qualitative results of video editing.

### C.2  Detailed quantitative results

Table 1 of the main paper reports video editing results when videos are split based on object instances. In addition, we conduct experiments with two splitting ways: (1) across different subjects, and (2) across different actions. The results are reported in Table 2. HOI-Swap consistently outperforms baseline approaches across different data splitting settings, demonstrating its effectiveness.

Furthermore, we provide a breakdown of video editing results in Table 3, separating results from in-domain videos (HOI4D) and out-of-domain videos (EPIC-Kitchens & TCN Pouring). We observe that the performance gain of HOI-Swap is higher for in-domain videos than out-of-domain videos. Future work could explore extending HOI-Swap's generalization capabilities.

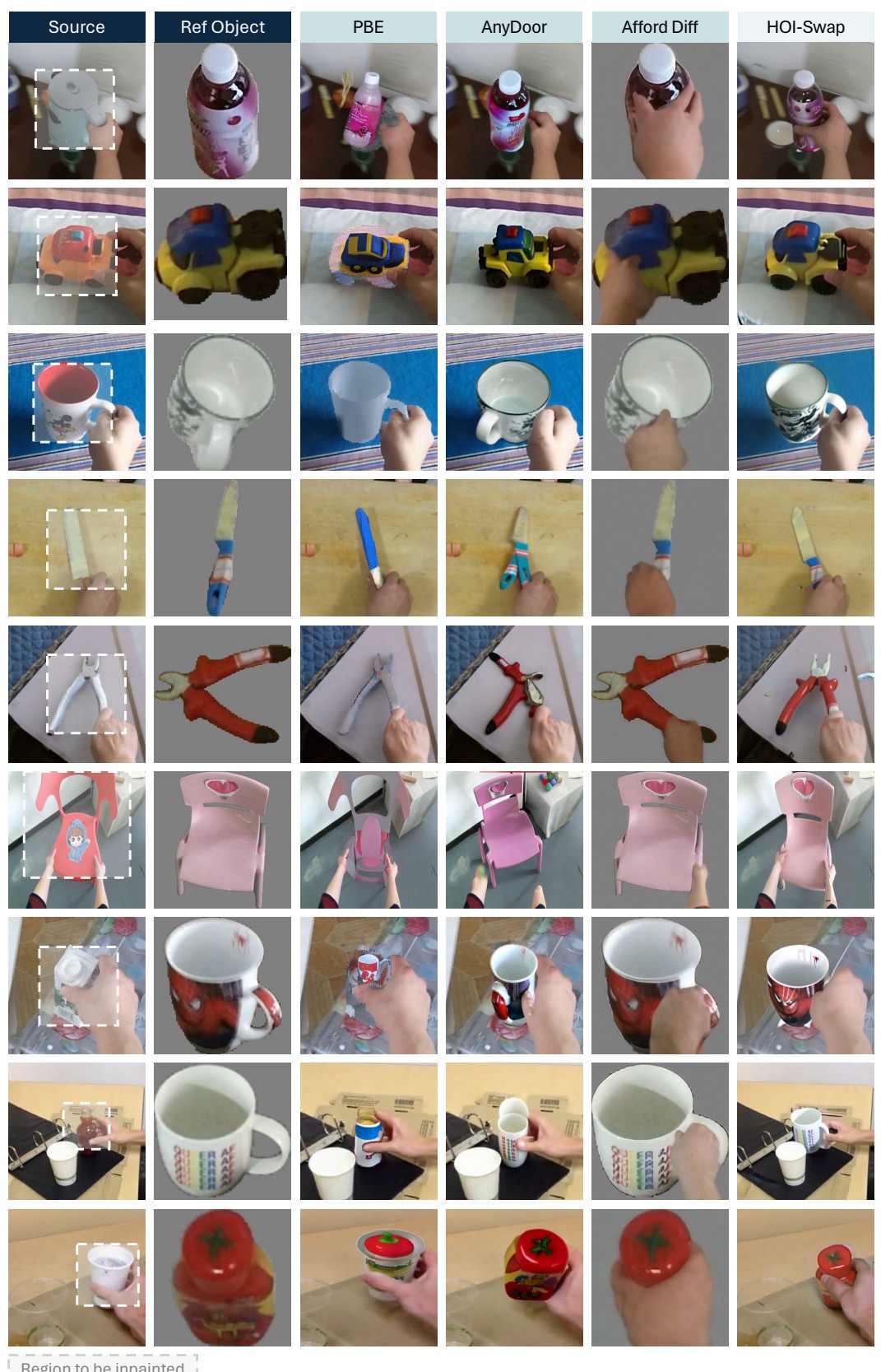

Figure 9: Qualitative image editing results of HOI-Swap on structured scenarios, with source images from HOI4D and TCN Pouring.

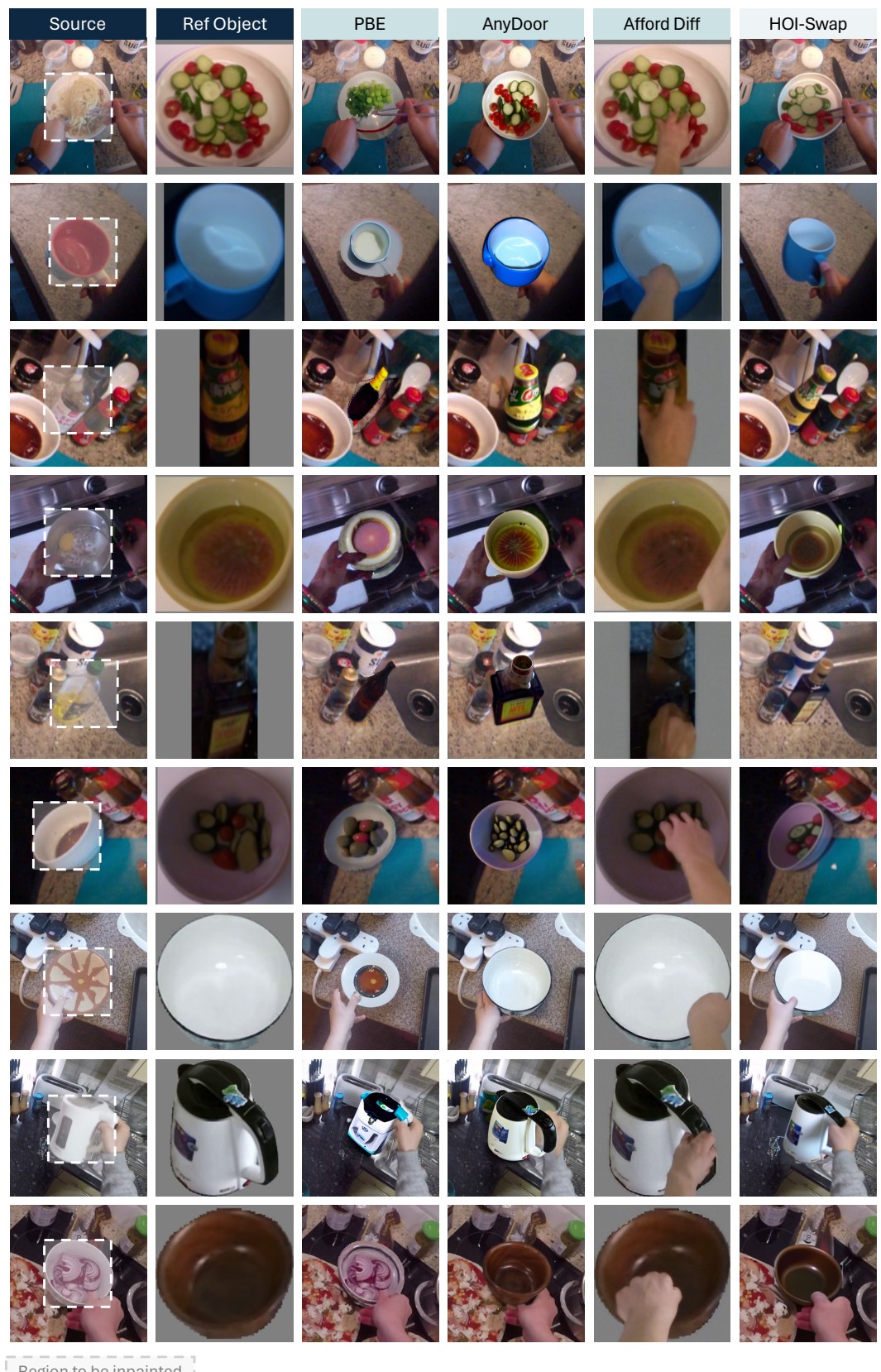

Figure 10: Qualitative image editing results of HOI-Swap on challenging in-the-wild scenarios, with source images from EgoExo4D and EPIC-Kitchens.

Table 2: Video editing results with splitting by subject and action (Table 1 in the main paper reports results with splitting by object instances).

| Method | split by subject | | | | | split by action | | | | |
|---|---|---|---|---|---|---|---|---|---|---|
| | subj. cons. | mot. smth. | cont. agr. | hand agr. | hand fid. | subj. cons. | mot. smth. | cont. agr. | hand agr. | hand fid. |
| Per-frame [8] | 85.6 | 95.2 | 73.6 | 63.3 | 62.9 | 84.5 | 94.7 | 82.8 | 57.6 | 71.0 |
| AnyV2V [29] | 67.2 | 95.3 | 68.2 | 27.7 | 10.0 | 69.8 | 94.7 | 74.6 | 30.6 | 19.9 |
| VideoSwap [18] | 88.0 | 97.4 | 71.8 | 29.6 | 69.0 | 87.2 | 97.1 | 87.7 | 39.8 | 79.4 |
| HOI-Swap (ours) | **90.0** | **98.2** | **80.4** | **84.4** | **90.3** | **89.6** | **97.8** | **93.3** | **77.4** | **94.6** |

Table 3: Video editing results breakdown: in-domain videos (left) and out-of-domain videos (right).

| Method | In-domain videos | | | | | | Out-of-domain videos | | | | | |
|---|---|---|---|---|---|---|---|---|---|---|---|---|
| | subj. cons. | mot. smth. | cont. agr. | hand agr. | hand fid. | user pref. | subj. cons. | mot. smth. | cont. agr. | hand agr. | hand fid. | user pref. |
| Per-frame [8] | 85.8 | 94.7 | 81.3 | 61.5 | 78.4 | 0.3 | 91.0 | 96.4 | 97.3 | 71.9 | 90.2 | 0.0 |
| AnyV2V [29] | 73.1 | 95.5 | 70.9 | 33.2 | 27.0 | 1.7 | 68.8 | 95.7 | 62.7 | 25.6 | 26.9 | 0.0 |
| VideoSwap [18] | 90.5 | 97.5 | 82.3 | 38.9 | 74.8 | 0.4 | 94.4 | 98.1 | 99.6 | **81.5** | 97.1 | 2.5 |
| HOI-Swap (ours) | **91.3** | **98.0** | **89.9** | **79.0** | **96.6** | **82.7** | **94.7** | **98.6** | **99.8** | 77.8 | **99.5** | **70.2** |

## C.3 User study analysis

Table 4 provides the breakdown of the evaluation dimensions for our HOI-Swap vs. baseline models on both image and video editing tasks. In terms of all metrics for both image and video editing, our HOI-Swap's editing results are consistently preferred by humans. Note that an exception is in object identity, where our HOI-Swap does not surpass Afford Diff. This is because Afford Diff focuses on synthesizing hands on fixed objects, directly copying and pasting the object image from the given reference object image.

Table 4: Breakdown of user study evaluation metrics. For image editing, we consider metrics including object identity, HOI realism, and overall quality. For video editing, we consider metrics including motion alignment and overall quality. Each value represents the frequency (expressed as a percentage) with which participants prefer that model considering a particular evaluation dimension. Standard deviations are calculated across participants to illustrate preference variability. For video editing, users were given an additional option to indicate if they found all edits unsatisfactory, which has an average rate of 10.9%. Note that our HOI-Swap **does not** surpass Afford Diff in the ability to preserve object identity. This is because Afford Diff focuses on synthesizing hands on fixed objects so the model will directly copy and paste the object image from the given reference object image.

| Image Editing | object identity | HOI realism | overall quality | Video Editing | motion alignment | overall quality |
|---|---|---|---|---|---|---|
| PBE [57] | $1.9 \pm 2.0$ | $11.4 \pm 7.1$ | $4.5 \pm 2.1$ | Per-frame | $0.2 \pm 0.8$ | $0.2 \pm 0.8$ |
| AnyDoor [7] | $40.3 \pm 11.1$ | $12.7 \pm 7.3$ | $15.6 \pm 6.5$ | AnyV2V [28] | $1.6 \pm 2.8$ | $1.2 \pm 2.7$ |
| Afford Diff [62] | $\mathbf{86.2 \pm 13.2}$ | $5.1 \pm 4.3$ | $7.6 \pm 6.8$ | VideoSwap [17] | $2.8 \pm 3.4$ | $1.2 \pm 2.0$ |
| HOI-Swap (ours) | $70.1 \pm 6.8$ | $\mathbf{71.5 \pm 9.1}$ | $\mathbf{72.1 \pm 10.8}$ | HOI-Swap (ours) | $\mathbf{84.5 \pm 10.3}$ | $\mathbf{86.4 \pm 8.6}$ |

## C.4 Ablation study

**DINO vs. CLIP encoder** Stage I and stage II of our pipeline employ the DINO and CLIP encoders, respectively, to align with their specific objectives. Stage I, focused on swapping the reference object within a single frame, benefits from the DINO encoder due to its enhanced ability to capture "objectness" compared with the CLIP encoder. The main emphasis of stage II is to transfer motion from the source video, and a CLIP encoder is adopted to provide scene context for generating the video. We additionally conduct experiments to explore the possibility of using a DINO encoder for stage II. As reported in Table 5, the two encoder variants perform similarly.

**Ground truth vs. estimated object masks** While it is generally assumed that users will provide the ground truth bounding box or segmentation mask of the object they wish to replace during

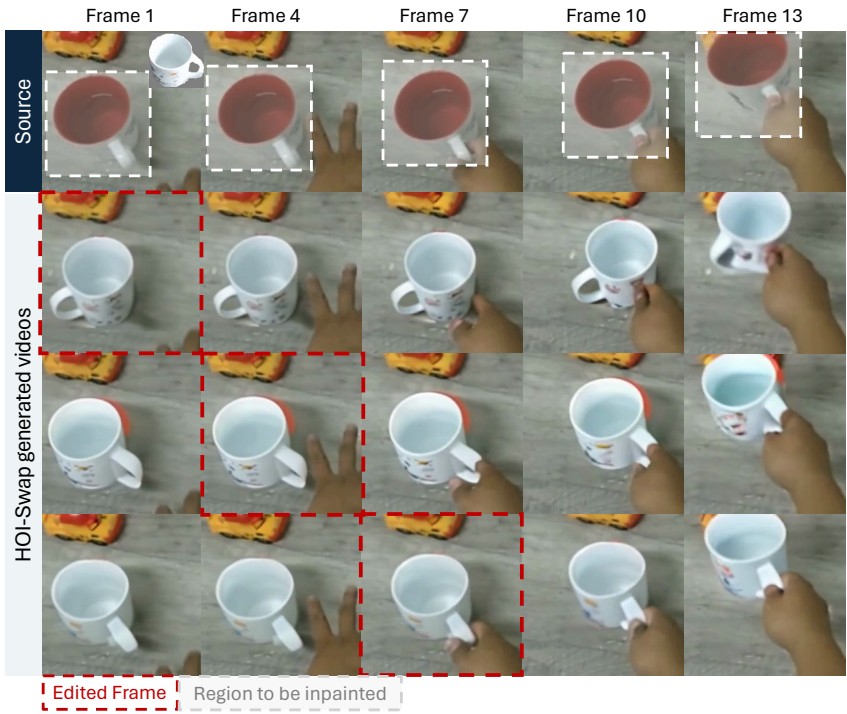

Figure 11: We compare generated results using different frames as the stage-I edit frame. Selecting the HOI contact frame (frame 7 in this example) for editing yields the best results.

Table 5: Video editing results: comparison of DINO and CLIP encoders for stage II. The two object encoders yield similar performance.

| Video Editing | subject consistency | motion smoothness | contact agreement | hand agreement | hand fidelity |
|---|---|---|---|---|---|
| HOI-Swap (DINO encoder) | 91.2 | 98.1 | 88.9 | 79.9 | 96.2 |
| HOI-Swap (CLIP encoder) | 91.4 | 98.0 | 89.9 | 79.0 | 96.6 |

inference [57, 7], we acknowledge the potential of incorporating automatic segmentation methods to reduce user effort. Following this, we applied SAM-2 [44] to identify bounding boxes on test videos as an alternative to manually providing ground truth. This method requires just a single click inside the object in the initial frame, followed by SAM-2 automatically tracking the target object. The quantitative comparisons are presented in Table 6. While there is some degradation across three HOI metrics, we believe this feature is valuable for downstream applications as it greatly eases the input requirements and improves user convenience. Note that the baseline approaches have more stringent input requirements than HOI-Swap, e.g. need precise object segmentation masks or additional text prompts. Even with the use of automatically generated masks, HOI-Swap demonstrates its great advantages over the baselines.

Table 6: Video editing results: comparison of using ground truth object bounding boxes vs. SAM-2 estimated ones as model input.

| Video Editing | subject consistency | motion smoothness | contact agreement | hand agreement | hand fidelity |
|---|---|---|---|---|---|
| Prior best | 90.5 | 97.5 | 82.4 | 61.5 | 78.4 |
| HOI-Swap (SAM-2 bbox) | 91.2 | 98.0 | 87.8 | 73.9 | 91.8 |
| HOI-Swap (GT bbox) | 91.4 | 98.0 | 89.9 | 79.0 | 96.6 |

**Masking strategy** In stage I, we mask the source frame with a square bounding box, as opposed to using the source object's segmentation mask as in [57]. This masking strategy not only directs

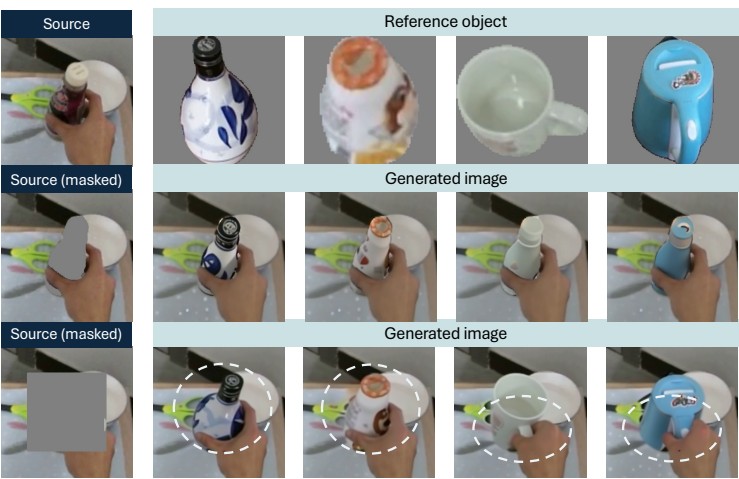

Figure 12: Comparison of two masking strategies. The stage-I model using a segmentation mask (row 2) only learns to fill the missing region without HOI awareness. In contrast, a square bounding box (row 3) also masks the hand-object interaction regions, requiring the model to reconstruct these interactions; this enables our stage-I model to effectively learn diverse HOI patterns from data.

the model to fill the predefined masked area but also to generate plausible hand-object interactions aligning with the source frame's hand position and the reference object. To demonstrate this point, we experimented with a variant of the stage I model that uses the original object's segmentation mask instead. As illustrated in Figure 12, this variant struggles with grasp changes when the reference and source objects differ, thereby reinforcing the effectiveness of our chosen approach.

**Editing frame selection**    One question naturally arising from our two-stage editing pipeline is determining which frame from the video sequence to edit. Recall that our training strategy (Section 3.3) prepares the stage-II model to be any-frame-conditioned, bringing maximum inference flexibility. Here, our key insight is to choose a hand-object contact frame for editing due to two main advantages: (1) Hand-object contact frames are particularly challenging as they require generating correct interaction patterns (e.g. grasps) compared with non-contact frames; prioritizing them ensures that the model can handle the most complex scenarios. (2) Hand-object contact frames offer the most comprehensive scene context for the stage-I model, aiding in spatial alignment.

To illustrate, in Figure 11, we compare video editing results when choosing the 1st, 4th and 7th frames in a sequence of picking up a mug. First, it is evident that the 7th frame is the most challenging to edit since it requires precise handling of the grasp region, while the 1st and 4th frame involve no or little hand interaction. Next, while single-frame edits for the 1st and 4th frame look reasonable (successfully swapping the mug), the lack of hand context in these frames leads to incorrect orientation of the mug handle. Consequently, this causes failures or imperfections in generating the subsequent frames 10 and 13, where the hand is unrealistically holding the mug.

**Motion points sparsity**    Following the discussion in Section 4.2 of the main paper, we present a comprehensive analysis of the impact of sampled point sparsity. Figure 13 illustrates two swapping cases, involving the motion of closing a trash can lid. For the left video, the two trash cans differ greatly in shape and closing mechanism. Using all points confuses the model, resulting in an incorrect lid-closing motion. Conversely, using no points generates a different motion of putting down the trash can rather than closing the lid. In this example, using 50% of the points yields the most plausible edit. For the right video, as discussed in the main paper, due to the similarities between the two objects, replicating the full motion by adopting 100% points is effective. In all, HOI-Swap allows users to flexibly decide the extent of motion transfer when generating content, accommodating different scenarios and object characteristics.

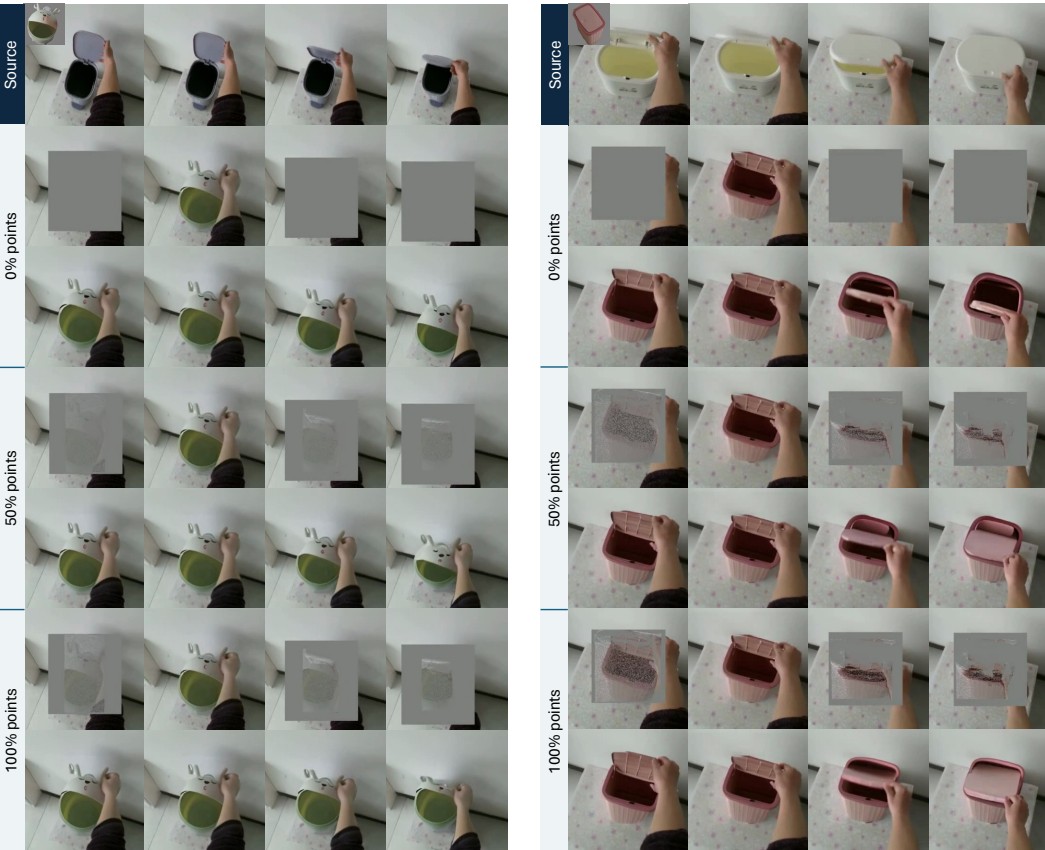

Figure 13: Ablation study of sampled motion points sparsity. The left figure illustrates a scenario where only partial motion transfer is desired, due to differences between the original and new object. The right figure showcases a scenario where full motion transfer is beneficial, owing to the similarities between the objects. We invite readers to view these examples in our project page.

## C.5 Discussion on sampling region

As described in Section 3.3, we perform uniform sampling of points within the bounding box region. When the new object differs greatly in shape from the original, sampled points might incorrectly capture motion—background points could be assigned foreground motion, and vice versa. A direct solution to mitigate this mismatch is to reduce the sampling density. Nevertheless, we find that HOI-Swap demonstrates considerable robustness even when a large number of points are sampled, adeptly managing potential discrepancies within the warped sequence $\mathcal{V}^{warp}$. As shown in Figure 14, with 50% points sampled, the visualized $\mathcal{V}^{warp}$ reflects the original object's shape to a certain extent, yet the model effectively distinguishes between background and the foreground object, recognizing the object as a whole and thereby generating plausible videos.

We identify two potential reasons for the model's effectiveness. First, our training strategy helps enhance the model's capability to discern between foreground and background elements, since we intentionally sample points within the bounding box region rather than solely within the object's segmentation mask. Consequently, the model is trained with a mix of in-object points and background points, preparing it to handle background points that may be sampled outside the new object region during inference. Additionally, the relative constancy of background and clear distinction between foreground and background contributes to the model's robustness, allowing it to recognize the object as a whole and accurately move it even when some regions inaccurately carry motion.

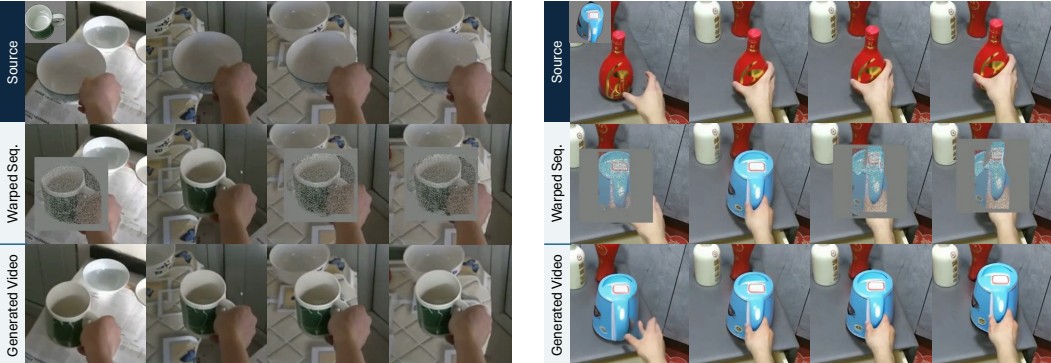

Figure 14: HOI-Swap demonstrates remarkable robustness, when object shapes differ greatly, leading to motion mismatches of sampled points. Even with a large sampled points density (50%), it effectively handles the flawed warped sequences $\mathcal{V}^{warp}$ and produces high-quality video edits.

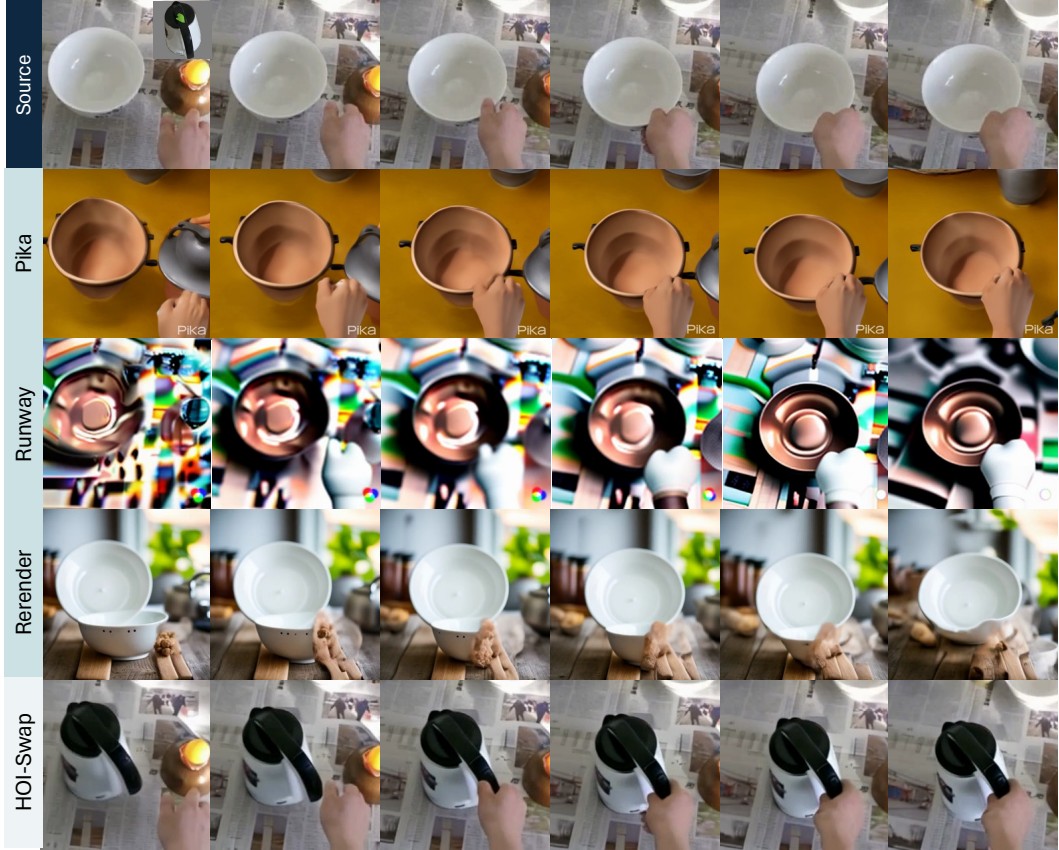

Figure 15: Comparison of HOI-Swap with text-guided diffusion models: Pika [30], Runway [10], and Rerender-a-video (Rerender) [59]. To evaluate these latest models on the object swapping task, we describe the reference object in text and prompt the models to replace the original object in the video. These approaches are unable to alter the shape of the bowl and fail to swap the original bowl with a kettle as required.

## C.6 Comparison with text-guided editing approaches

To understand how the latest text-guided diffusion models perform on this task, we provide qualitative comparison of HOI-Swap with three models: Pika [30], Runway [10], and Rerender A Video [59]. We adopt the text prompt: "change the object in the video to a kettle while keeping background and

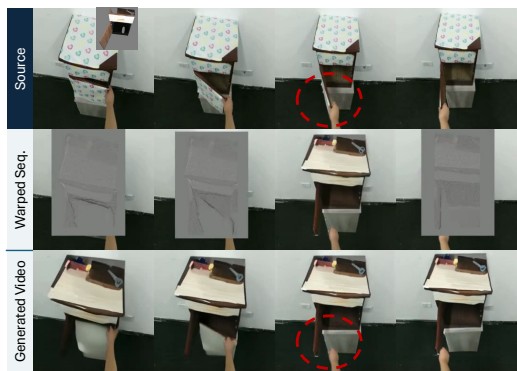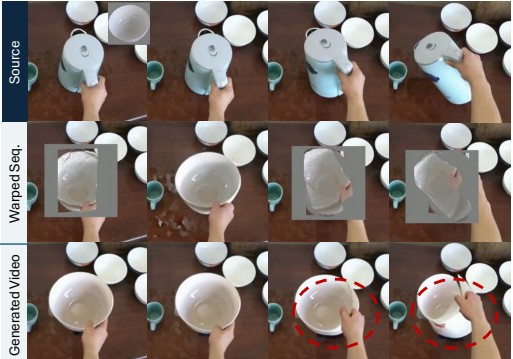

Figure 16: Failure cases of HOI-Swap. Left: stage-I failure, Right: stage-II failure.

motion the same", which proved most effective compared to more complex prompts. As shown in Figure 15, these approaches edit the video in a way that preserves the original object shape, failing to generate the new object or adapt the HOI patterns. These results underscore the unique capabilities of HOI-Swap.

## D  Limitations

One area for improvement stems from the sampling region discussion in Section C.5, with sampled points potentially carrying wrong motion when object shape changes, particularly when a large number of points are sampled. To further refine our model's performance, a future direction is to explore fine-grained spatial control in motion alignment. Specifically, concentrated sampling could provide the model with the precise ability to modulate specific regions of an object. These enhancements are anticipated to augment the model's adaptability and accuracy in editing complex HOI sequences.

Another direction is the development of an automated method to determine the sparsity of motion points based on the object swapping scenario and the motion complexity in the original video. This would allow the model to autonomously decide the extent of motion transfer necessary when generating new content.

In addition, we present failure cases in Figure 16. The left example shows a stage-I failure, where HOI-Swap fails to achieve accurate spatial alignment. The right example illustrates a stage-II failure, where noise in the warped sequence impedes model's understanding of the reference object, leading to wrong motion patterns.

Finally, we emphasize that the problem of swapping in-contact objects in videos poses significant challenges, particularly with complex HOI sequences. We recognize the limitations of HOI-Swap in its current stage and view this work as an initial step towards resolving these challenges. Looking ahead, we aim to incorporate more precise spatial and motion controls, and expand HOI-Swap's capabilities to handle longer video sequences with more intricate HOIs.

## E  Societal Impact

This project aims to equip the community with a powerful tool for swapping customized objects into videos, particularly improving the realism of hand-object interactions. However, there is a potential risk that malicious users could misuse this technology to produce deceptive videos involving real-world individuals, potentially misleading viewers. This issue is not exclusive to our approach but is a common concern across various video editing and generating modeling techniques. To mitigate such risks, one possible solution is to introduce subtle noise perturbations to published images to disrupt the customization process. Furthermore, implementing invisible watermarking in the generated videos could help prevent misuse and ensure proper attribution of the content.

