# OpenReview forum: "HOI-Swap: Swapping Objects in Videos with Hand-Object Interaction Awareness"
_NeurIPS.cc/2024/Conference — NeurIPS 2024 poster_

### Official Review · Reviewer_rdgi · 2024-07-09

**Soundness:** 3
**Presentation:** 3
**Contribution:** 3
**Rating:** 6
**Confidence:** 4

**Summary:**

This paper presents HOI-Swap, a diffusion-based video editing framework for object swap editing. HOI-Swap involves two stages. In the first stage, the authors train an image-editing model to swap object in one frame. In stage II, the authors first warp a video from the edited frame using the optical flow sampled from the source video. Then a video diffusion model is trained to reconstruct the whole video from the warped frame sequences. Through this two-stage approach, HOI-Swap successfully addresses three main challenges in HOI video editing: (a) HOI-aware capabilities, (b) spatial alignment of the object with the hand, and (c) temporal alignment with the source video. Experiments demonstrate that HOI-Swap outperforms both image-editing and video-editing baselines.

**Strengths:**

1. This paper address a novel task: HOI video editing.
2. This paper pinpoints the three main challenges facing this task.
3. The video-editing stage employs optical flow from the source video to achieve temporal alignment with the source video.
4. HOI-Swap demonstrates better performance than existing video-editing approaches in HOI video editing.

**Weaknesses:**

1. Stage I does not seem to address HOI-awareness explicitly. The authors should explain why the image-editing model can perceive HOI.
2. The authors did not mention the generalizability of their method across unseen kinds of objects.

**Questions:**

1. How can the image-editing model in stage I perceive HOI? The authors should provide more explanation and maybe some more ablation studies to support their results.
2. The whole pipeline seems applicable to general object-swap tasks. Have the authors tested the model on other objects?
3. Are the backgrounds in the test set all seen by the model during training?

**Limitations:**

The authors have adequately addressed the limitations.

---

> ### Author Rebuttal · Authors · 2024-08-06
>
> We thank reviewer rdgi for the helpful comments and for providing thoughtful feedback on our work.
>
> ---
>
> **1. HOI awareness in stage I**
>
> > How can the image-editing model in stage I perceive HOI?...maybe some more ablations
>
> The model’s ability to perceive HOI fundamentally relies on the data it is trained on. By providing the model with a large volume of HOI images, it learns the diverse ways in which hands interact with various objects. HOI-rich data acts as the driving force, equipping the model with the necessary insights to accurately replicate and understand these interactions.
>
> In terms of the specific techniques, in stage I, we mask the source frame with a square bounding box (Ln 162-166), as opposed to using the source object’s segmentation mask as in [57]. This masking strategy not only directs the model to fill the predefined masked area but also to generate plausible hand-object interactions aligning with the source frame’s hand position and the reference object. To demonstrate this point, we experimented with a variant of the stage I model that uses the original object's segmentation mask instead. As illustrated in Figure R2 of the rebuttal PDF, this variant struggles with grasp changes when the reference and source objects differ, thereby reinforcing the effectiveness of our chosen approach. We will incorporate this discussion in the paper.
>
> ---
>
> **2. Applicability to general object swaps, testing on other objects?**
>
> ​​Yes. As noted in Ln 567-568, we designed our train-test split based on object instances, to evaluate our model’s ability to generalize to unseen object instances, such as a new mug not encountered during training. Our qualitative results indicate that HOI-Swap can handle novel object instances and deliver good edits.  Moreover, as demonstrated in Figures 4 and 8 of the paper, HOI-Swap performs robustly in general object-swap tasks, in cluttered scenes where no hand is present, showcasing its strong capability across diverse swapping scenarios.
>
> On the other hand, object category-level generalization requires the model to generate plausible interactions with a completely unseen object category (e.g., accurately depicting a hand holding scissors, despite the model never encountering scissors during training), is considerably more challenging and not validated here. This requires the model to acquire a broader understanding of novel interactions. We acknowledge this as an exciting direction for future work.
>
> > The whole pipeline seems applicable to general object-swap tasks. Have the authors tested the model on other objects?
>
> Following the discussion above, our experimental setup aims at assessing generalization across new object instances. If the reviewer could specify or elaborate on “other objects,” we would be happy to address this further.
>
> ---
>
> **3. Background generalization**
>
> > Are the backgrounds in the test set all seen by the model during training?
>
> We provide zero-shot video editing results on two datasets (EPIC-Kitchens and TCN Pouring), where the test videos feature backgrounds not encountered by the model during training. This showcases HOI-Swap’s generalization ability to new backgrounds.

---

> > ### Comment · Reviewer_rdgi · 2024-08-12
> >
> > Thank the authors for their response. I think most of my concerns are addressed.
> >
> > I want to clarify Question 2. By "general object-swap tasks" I mean object-swap tasks without HOI and "other objects" means different categories of objects. The authors have addressed my concerns on object category-level generalization. I hope the authors include more discussion of the general object-swap task in final manuscript.
> >
> > Generally speaking, this is an interesting and novel paper. For now, I will keep my initial score.

---

> > > ### Author Response · Authors · 2024-08-12
> > > **Response to Reviewer rdgi's further comment**
> > >
> > > Thank you for your insightful comments and for clarifying your question regarding general object-swap tasks. In the experiments, we indeed include a diverse set of images to evaluate our model comprehensively. As noted in Ln 259-261, our evaluation involves both HOI-contact and non-HOI-contact scenarios:
> > >
> > > + Our editing benchmark comprises 5,000 test images, of which 20.1% represent non-HOI-contact scenarios, highlighting our model's ability to handle general object-swap tasks where hand-object interaction is absent.
> > > + For qualitative results, row 3 and 4 of Figure 4 in the main paper, row 3, 5 and 6 of Figure 9 in Supp. showcase general swapping scenarios without human hands, emphasizing our model's versatility across different contexts.
> > >
> > > We appreciate your suggestion and will expand on this discussion in the updated manuscript. Thank you once again for your thoughtful review.

---

### Official Review · Reviewer_gbKB · 2024-07-11

**Soundness:** 3
**Presentation:** 2
**Contribution:** 2
**Rating:** 6
**Confidence:** 4

**Summary:**

This work presents a novel approach for object insertion in hand-object interaction scenarios. The proposed approach consists of two stages: image-based editing to for precise alignment of hand with the inserted object, and video-based editing for motion alignment with the original video. The model is trained in a self-supervised manner and also imparts a varying level of controllability to adjust the degree of motion alignment based on object changes. Extensive experiments on HOI4D, EgoExo4D, EPIC-Kitchens and TCN Pouring datasets show the effectiveness of the proposed approach in various scenarios.

**Strengths:**

- The proposed 2-stage approach to decompose the complexity of the task is intuitive and handles the challenges of hand-object interaction changes, correct object placement and temporal alignment of HOI motion effectively.
- The first stage can also be used in a stand-alone manner for image editing tasks and the controllability in motion alignment is a useful feature.
- The self-supervised training strategy circumvents the need for collecting paired training data for this task..
- Extensive experiments (Tab.1) on HOI4D & EgoExo4D show the effectiveness of the proposed approach over several baselines for both image and video editing tasks. The zero-shot generalization setting to EPIC-Kitchens and TCN Pouring datasets are also considered.
- Visualizations in Fig.4,5 are helpful in understanding the capabilities of the proposed approach.

**Weaknesses:**

- Several details about the experimental setup are missing from the main paper. These details are important to understand the scope of the claims in the paper. While the supplementary contains more details, it'd be helpful to have more clarity on the following aspects:
    - For evaluation on HOI4D & EgoExo4D, how is the test split created? Do the held-out videos contain different object categories or different instances from the same category or different subjects performing the experiments or different actions being performed? Having quantitative experiments in these different settings would help in understanding the benefits and limitations of the proposed approach.
    - Do the results in Tab.1 span all 4 datasets? Again, it'd be helpful to have a breakdown of the results in different settings to understand where the proposed approach is most effective.
- Are all the baselines trained in the same setup as the proposed approach? I understand that it might not be feasible to implement/train all baselines due to differences in architecture and compute requirements, but it'd be useful to clearly describe the protocols followed for each baseline. Are these baselines retrained or used in a zero-shot manner? For example, AffordDiff [62] only takes a single RGB image of the object as input and generates the HOI image, it does not take the reference video into account so it is expected that the object orientation may not be consistent with the video (L43-50). AffordDiff also allows for controllability in the layout & orientation of the generated hand so it can be modified to match the hand in the reference video. I'm not asking to reimplement AffordDiff to adapt to this new setting, but it'd be useful to have these details in the main paper to understand the differences between baselines and the proposed approach.
- While the 2-stage decomposition of the approach makes sense, it'd also be useful to quantitively verify that it is indeed better than the 1-stage analog. For example, in Fig. 3, Stage-2 can take the DINO encoding of the reference object instead of the generated image from Stage-1 to create a 1-stage approach. This is not required for rebuttal, but a suggestion to validate the 2-stage approach.

**Questions:**

I need clarifications on the following aspects to better understand the results (more details in the weaknesses above):
- For evaluation on HOI4D & EgoExo4D, how is the test split created? Do the held-out videos contain different object categories or different instances from the same category or different subjects performing the experiments or different actions being performed?
- Do the results in Tab.1 span all 4 datasets? It'd be helpful to have a breakdown of the results in different settings to understand where the proposed approach is most effective.
- Are all the baselines trained in the same setup as the proposed approach? It'd be useful to clearly describe the protocols followed for each baseline. Are these baselines retrained or used in a zero-shot manner?

**Limitations:**

L330-332 in the main paper defers the limitations and failure modes to the supplementary without providing any insights. Please add these details to the main paper.

---
I have read all the reviews and the rebuttal. I thank the authors for providing additional clarifications, which help me understand the paper better and I have increased my score to Weak Accept.

---

> ### Author Rebuttal · Authors · 2024-08-06
>
> We thank reviewer gbKB for the helpful comments and for providing thoughtful feedback on our work.
>
> ---
>
> **1. Clarification on experimental setup**
>
> ***(i) Data split***
>
> As we focus on the problem of “object” editing, videos are split based on object instances (Ln 567-568). The held-out videos feature different instances of the same object categories, testing the model’s ability to swap unseen instances (e.g., a different mug not seen during training) rather than entirely new categories (e.g., scissors, if scissors were not included in the training data). Addressing object category-level generalization presents a significantly more challenging problem, requiring the model to generate plausible interactions with a completely unseen object category (e.g., accurately depicting a hand holding scissors, despite the model never encountering scissors during training). We acknowledge this as an exciting direction for future work.
>
> Inspired by your comment on other data splitting ways, we conducted experiments with two new splits: (1) across different subjects, and (2) across different actions. The results are reported in Table R1 in the rebuttal PDF. HOI-Swap consistently outperforms baseline approaches in these new settings, demonstrating its effectiveness.
>
> ***(ii) Baselines implementation***
>
> Baselines including PBE [57], AnyDoor [8] and AnyV2V [29] are adopted in a zero-shot manner. Afford Diff [62] utilizes the same training data source (HOI4D) as ours. We do not provide additional layout or hand orientation information to Afford Diff. VideoSwap [18], a one-shot-tuned video editing approach, is trained on the source video using the default setup in their official repository.
>
> Note that we also evaluate on two out-of-domain datasets (EPIC-Kitchens and TCN pouring), where all approaches (except VideoSwap which is tuned on the source video), including ours, are applied in a zero-shot manner.
>
> We appreciate your feedback and will ensure these clarifications are included in the main paper.
>
> ***(iii) Results breakdown***
>
> As suggested, in Table R2 of the rebuttal PDF, we provide a breakdown of video editing results, separating results from in-domain videos (HOI4D) and out-of-domain videos (EPIC-Kitchens & TCN Pouring). Note that the video editing results in Table 1 span three datasets (HOI4D, EPIC-Kitchens and TCN Pouring) with EgoExo4D excluded due to its absence of high-FPS mask annotations (see Ln 575-577). We observe that the performance gain of HOI-Swap is higher for in-domain videos than out-of-domain videos. We will update our manuscript to incorporate these discussions.
>
> ---
>
> **2. Validation of the two-stage approach**
>
> We appreciate the suggestion to verify the efficacy of our two-stage design compared to a one-stage counterpart (though the reviewer says it’s not necessary for rebuttal) and have conducted additional experiments to assess this. We train one VideoLDM [2] that takes the reference object image and the masked frame sequence as input and outputs the edited video; this serves as a trained baseline for a one-stage object swapping approach.
>
> We provide both qualitative and quantitative comparisons in Figure R1 of the rebuttal PDF. The one-stage approach does not yield satisfactory results, specifically failing to preserve the reference object’s identity. We believe this deficiency stems from the model’s need to handle both spatial and temporal alignments simultaneously. We will add this analysis in the paper.
>
> ---
>
> **3. Inclusion of limitations in main paper**
>
> We will add a summary of the key limitations and failure modes (currently discussed more in Supp. and Supp. video) directly into the main text to ensure comprehensive accessibility of these details within the main paper.

---

> > ### Comment · Reviewer_gbKB · 2024-08-13
> > **Thanks for the clarifications**
> >
> > I appreciate all the clarifications provided by the authors. I have increased the score to Weak Accept and suggest to include these additional details in the final version.

---

> > > ### Author Response · Authors · 2024-08-13
> > >
> > > Thank you for your insightful feedback and valuable support! We are pleased to confirm that we have fully addressed your concerns. We will make the necessary updates to the manuscript in light of your thoughtful suggestions.

---

### Official Review · Reviewer_NtRs · 2024-07-12

**Soundness:** 3
**Presentation:** 2
**Contribution:** 2
**Rating:** 5
**Confidence:** 4

**Summary:**

This article focuses on proposing a two-stage network HOI-Swap, a video editing framework designed for precise object edits with HOI awareness.To address the problem of real perception of HOIs as well as spatial and temporal alignment with the original video, HOI- Swap first stage focuses on solving HOI awareness and establishing spatial alignment by training an image restoration diffusion model to swap objects in one frame. The second phase deforms video sequences from edited frames by tracking inter-frame points with randomly sampled optical flow, and trains a video diffusion model to generate new videos.

**Strengths:**

1. This paper is well organized and easy to understand.
2. The ablation experiment fully confirms the effectiveness and contribution of the improvement measures proposed in this paper.

**Weaknesses:**

1. Limited innovation. The methodology of this paper is a combination of existing work and is more like a technical report than an academic paper.
2. Lack of dynamic HOI awareness transfer. The second stage does not embody HOI interaction awareness. The single-frame propagation only propagates the motion information of the object and does not embody how the HOI-aware information is conveyed in the video sequence.
3. Insufficient experiments. This paper only demonstrates HOI awareness in single-frame image restoration, and in the video aspect, either the scene with little change in hand poses or the scene with little change in object characteristics is demonstrated, but for the complex video sequences with large changes in object characteristics and the need for dynamic changes in hand pose, the paper does not give the relevant effects to demonstrate.

**Questions:**

1.line 174, this paper suggests that data augmentation techniques such as flipping, rotating, perspective transformations, etc. are performed due to the difference in size and pose of the target and reference objects, so why not use scaling techniques to bridge the gap between large and small objects?
2.Line205, this paper proposes to apply the largest bounding box as mask to the whole video sequence, so for the case of a relatively large range of object movement, will a larger mask range lead to the proportion of sampling points that does not truly reflect the proportion of extracted object features and motion information？(For example, if the target object only occupies 1/4 size of the mask in a certain frame, then a uniform sampling ratio of 50% is also difficult to extract the object motion information).
3. Why not continue the use of DINO for the first stage network instead of switching to CLIP in the second stage network?

**Limitations:**

Yes

---

> ### Author Rebuttal · Authors · 2024-08-06
>
> We thank reviewer NtRs for the helpful comments and for providing thoughtful feedback on our work.
>
> ---
>
> **1. Limited innovation**
>
> > The methodology of this paper is a combination of existing work
>
> We respectfully disagree. As acknowledged by three other reviewers, our paper presents both a novel task and approach. Reviewer ih2o noted that “The HOI-Swap method is novel and addresses a yet unexplored problem”, reviewer gbKB mentioned it “presents a novel approach”, and reviewer rdgi recognized it as addressing “a novel task: HOI video editing”.
>
> ***Clarifications of innovations.*** In the related works section, we thoroughly discuss existing approaches and outline their differences with us. Our experimental results (Ln 316-321) further demonstrate that the leading generative methods fall short in producing satisfactory edits within HOI scenarios, thus underlining the importance and necessity of HOI-Swap.
>
> ***Innovation highlights.*** (i) Problem Space: We investigate HOI-aware object swapping in videos, an area where current leading generative models falter (see Figure 4 of the main paper). This problem space has not been previously explored, and our work serves to fill this gap in the field. (ii) Technical Contributions: Our proposed editing framework simplifies complexity by dividing the task as two stages. The first stage introduces HOI awareness and spatial alignment—elements notably missing in existing image editing methods. The second stage offers controllable motion alignment with the original video, utilizing motion points and warping techniques; this approach stands in stark contrast to existing video editing approaches that enforce 100% motion alignment with shape-based conditional signals.
>
> We would appreciate it if reviewer NtRs could identify any potentially overlooked references, allowing us to further clarify and set apart our contributions.
>
> ---
>
> **2. Clarification on HOI awareness**
>
> > The second stage does not embody HOI interaction awareness. The single-frame propagation only propagates the motion information of the object and does not embody how the HOI-aware information is conveyed in the video sequence.
>
> We appreciate your comment and clarify below. HOI-awareness is a spatial property that can be reliably captured in a static frame and generally remains consistent over time.  In our study, we observe that across the four datasets we use, HOI interactions are stable throughout the sequence. As an intuitive example, when swapping a hand-held mug, one HOI frame suffices to conceptualize how a new object would replace the original, including any necessary adjustments in hand grasp patterns; we then propage this edit to the remaining frames using our controllable motion guidance techniques.
>
> Our two-stage design is driven by this principle. We recognize the reviewer's concern regarding scenarios where an object may undergo multiple distinct hand interactions within a short clip, though these are uncommon (based on our observation across 4 datasets). As noted in Ln 741-745, we view our work as an initial step towards HOI-aware video editing challenges, and plan to explore more complex video sequences in future work.
>
> ---
>
> **3. Complex video sequences with large object changes**
>
> > Either the scene with little change in hand poses or the scene with little change in object characteristics is demonstrated, but for the complex video sequences with large changes in object characteristics and the need for dynamic changes in hand pose, the paper does not give the relevant effects to demonstrate
>
> Through various qualitative generation results in our paper and Supp. video, we show that HOI-Swap can adeptly address diverse object swaps in videos. For instance, Figure 1 of the main paper illustrates the replacement of a kettle with a bottle and a bowl—objects that greatly differ in shape and appearance from the original kettle. Figure R2 of the rebuttal PDF (first and last row) provides further evidence of its editing capabilities for differently-shaped objects.
>
> Moreover, our Supp. video includes scenarios featuring various object and action variations. Object variation examples include swapping a bottle with a mug (page 5), a bowl with a kettle (page 5) and a trash can with another differently-shaped one (page 9-10). For action variations, examples include closing a laptop display (page 4), picking up a scissor (page 4), uprighting a tilted bottle (page 5), pushing and rotating toy car (page 6), and closing trash can lid (page 9-10).
>
> We invite the reviewer to provide more details on what they consider “complex video sequences with large object changes,” so we can address this aspect more thoroughly. Lastly, as noted in Ln 316-321, HOI-aware video editing is a very challenging problem, and our experiments reveal that even leading video generation approaches struggle with the scenarios we have tested. We acknowledge in our limitations statement (Ln 744-745) that improving HOI-Swap’s capabilities to handle longer video sequences with more intricate HOIs is an important future direction.
>
> ---
>
> **4. Clarification on scaling**
>
> > Why not use scaling techniques…?
>
> We indeed incorporate scaling as our data augmentation by randomly resizing the reference object (Ln 644-646 of Supp.). We'll ensure to clearly mention this in Ln 174 of the main paper.
>
> ---
>
> **5. Clarification on bounding box**
>
> The largest bounding box is used only for preparing the masked input sequence (Ln 205) to prevent exposing the source object to the model, while sampling points sparsity is decided by the conditioning frame’s bounding box (Ln 222). We'll clarify that these are separate processes in the main paper.
>
> ---
>
> **6. Motivation of DINO vs. CLIP encoders**
>
> Please refer to our general response.

---

> > ### Comment · Reviewer_NtRs · 2024-08-14
> >
> > Thanks to the authors for providing a rebuttal. The authors' replies on scaling as a data augment technique and mask region selection as well as additional experiments evaluating CLIP vs DINO as an encoder for stage 2 answered my confusion.
> >
> > 1) Novelty.  I agree that this paper primarily focuses on a new task: video HOI editing. However, when it comes to the more significant aspect of HOI awareness, I still believe that there isn't much novel design.
> > 2) Dynamic HOI Awareness. As written in the rebuttal, HOI-Swap lacks the ability to handle long video sequences with more complex HOI variations, which is an important direction for video editing.
> > 3) Object Shape Variations. I am more interested in examples where the hand-object interaction pose varies greatly due to different categories. I agree with the validity of replacing the kettle with a bottle and a bowl-like object in Fig. 1 of the main paper, but the poses of the grasped objects are similar.
> >
> > I agree with Reviewer ih2o that this paper should clarify its downstream scenarios and further explain its capability boundaries.
> > Considering that the authors solved some of my confusion and the paper does propose a new task, I raised the score to Borderline Accept.

---

### Official Review · Reviewer_ih2o · 2024-07-12

**Soundness:** 3
**Presentation:** 3
**Contribution:** 3
**Rating:** 7
**Confidence:** 3

**Summary:**

The work considers the task of swapping the objects in ego-centric short clips with hand-object interactions. The manuscript claims to introduce this sub-task within the field of generative video editing. Concretely, HOI-Swap starts from RGB video, object area bounding box, and an image of the target object, and generates a new video, transferring hand movement to the new object, possibly adapting to the changing functionality of the object. The method has two stages: inpainting a single frame, and extending it to the whole sequence ensuring consistency. HOI-Swap is thoroughly evaluated against recent video and image editing methods, also the work demonstrates qualitative examples of generalization to novel datasets.

**Strengths:**

1) The HOI-Swap method is novel and addresses a yet unexplored problem;
2) The proposed method is technically sound. It employs several techniques to ensure robustness and generalization of the resulting model: reference object augmentation technique to ensure robustness to viewpoint orientation, robust masking (square mask, consistent across frames to prevent overfitting), and randomized selection of the anchor frame.
3) Another notable feature of the method is its controllability through motion guidance. This original idea enables to variation of the amount of preserved information from the source object's motion to the target, thus allowing the method to adapt to the changing functionality of the objects.
4) The manuscript is well-written and easy to follow.

**Weaknesses:**

A minor weak point is the lack of detailed discussion on downstream applications of the model. The Introduction section touches on this topic, however, an example of possible demos or a deeper discussion of possible usages of the model could help to better motivate the importance of the introduced task and the proposed method.

Another possible area for improvement is automatic input mask annotation. Currently, the method requires a bounding box of the object as an input, however, with current advancements in the segmentation methods this requirement can be alleviated (or at least quantitatively compared to the GT bbox usage).

**Questions:**

1) Stages 1 and 2 employ different image encoders (DINO and CLIP), what is the motivation for this choice?
2) Lines 212-220 discuss the ability to vary the number of control points sampled for motion guidance. What is the default value used for quantitative evaluations?

**Limitations:**

A detailed discussion of limitations is present in the Appendix with complementary examples in a supplementary video.

---

> ### Author Rebuttal · Authors · 2024-08-06
>
> We thank reviewer ih2o for the helpful comments and for providing thoughtful feedback on our work.
>
> ---
>
> **1. Downstream applications**
>
> ***(i) Entertainment.*** As showcased in Figure 1 of the main paper and Supp. video (pages 2-3), HOI-Swap can be applied in scenarios where object modification is required without reshooting footage. For example, in advertising, there may be situations where a pre-recorded video needs to adapt to new sponsorship requirements by replacing a soda can in the video with a water bottle. HOI-Swap offers a practical tool to seamlessly swap the objects, while maintaining realistic hand interaction patterns that adapt to the new object's shape and affordance.
>
> ***(ii) Data augmentation in robotics.*** There is an increasing trend of teaching robots through videos of humans performing tasks (Ln 24-26). By recording just a single video of picking up a mug, HOI-Swap can generate multiple variations of this video with different objects (e.g., bottles, bowls, kettles), all following the same motion trajectory. This capability can greatly reduce the need for extensive data collection in robotics.
>
> ---
>
> **2. Automatic input mask annotation**
>
> > Currently, the method requires a bounding box of the object as an input, however, with current advancements in the segmentation methods this requirement can be alleviated (or at least quantitatively compared to the GT bbox usage).
>
> We appreciate the suggestion. While it is generally assumed that users will provide the ground truth bounding box or segmentation mask of the object they wish to replace during inference [8, 57], we acknowledge the potential of incorporating automatic segmentation methods to reduce user effort. Following this, we applied the recently released SAM-2 to identify bounding boxes on test videos as an alternative to manually providing ground truth. This method requires just a single click inside the object in the initial frame, followed by SAM-2 automatically tracking the target object. The quantitative comparisons are presented in the table below. While there is some degradation across three HOI metrics, we believe this feature is valuable for downstream applications as it greatly eases the input requirements and improves user convenience. Note that the baseline approaches have more stringent input requirements than ours, e.g. need precise object segmentation masks, additional text prompts (see Ln 590-596). Even with the use of automatically generated masks, HOI-Swap demonstrates its great advantages over the baselines. We will add this discussion in the paper.
>
> | Model    | CLIP consistency | Motion smoothness | Contact agreement | Hand mIOU | Hand confidence |
> |-------------|:----------------:|:-----------------:|:-----------------:|:---------:|:---------------:|
> | Prior best	                    | 90.5             | 97.5              | 82.4              | 61.5      | 78.4            |
> | HOI-Swap (SAM-2 bbox)    | 91.2             | 98.0              | 87.8              | 73.9      | 91.8            |
> | HOI-Swap (GT bbox)          | 91.4             | 98.0              | 89.9              | 79.0      | 96.6            |
>
> ---
>
> **3. Motivation of DINO vs. CLIP encoders**
>
> > Stages 1 and 2 employ different image encoders (DINO and CLIP), what is the motivation for this choice?
>
> Please refer to our general response.
>
> ---
>
> **4. Default sampling points sparsity for motion guidance**
>
> The default value used for quantitative evaluations is 50%. We appreciate your pointing it out and will include this information in the paper.

---

> > ### Comment · Reviewer_ih2o · 2024-08-12
> > **Reply by Reviewer ih2o**
> >
> > I thank the authors for providing the rebuttal. The additional experiments (evaluation of CLIP vs DINO as an encoder for stage 2 and evaluation of using non-GT mask predicted on the fly) clarify my concerns and strengthen the work.
> >
> > After considering the authors' replies and other reviews, I have decided to increase my score to accept. The work tackles a novel problem, providing a solid baseline for future research.

---

> > > ### Author Response · Authors · 2024-08-13
> > >
> > > Thank you for your insightful feedback and valuable support! We are pleased to confirm that we have fully addressed your concerns. We will make the necessary updates to the manuscript in light of your thoughtful suggestions.

---

### Author Rebuttal · Authors · 2024-08-06

We thank all reviewers for their thoughtful and constructive review of our manuscript.

Three of the four reviewers recommend accepting. We were encouraged that the reviewers found our problem and approach novel (ih2o, gbKB, rdgi), technically sound (ih2o), and that our paper pinpoints and addresses this task’s challenges effectively (rdgi, gbKB), backed up with extensive experiments (gbKB), ablations (NtRs) and helpful visualizations (gbKB).

In response to the feedback received, we provide a general response here to address a query shared by two reviewers (ih2o and NtRs), and individual responses below to specific points from each reviewer. Please refer to the attached rebuttal PDF, where supplementary figures and tables have been provided to substantiate our responses.


---

**Motivation of DINO vs. CLIP encoders**


We thank Reviewer ih2o and reviewer NtRs for raising this question. Stage I and stage II of our pipeline employ the DINO and CLIP encoders, respectively, to align with their specific objectives. Stage I, focused on swapping the reference object within a single frame, benefits from the DINO encoder due to its enhanced ability to capture “objectness” compared with the CLIP encoder. The main emphasis of stage II is to transfer motion from the source video, and a CLIP encoder is adopted to provide scene context for generating the video. Given the reviewers’ query, we additionally conduct experiments to explore the possibility of using a DINO encoder for stage II. As reported in the table below, both encoder variants perform similarly. We will add this discussion in the paper.

|                          | CLIP consistency | Motion smoothness | Contact agreement | Hand mIOU | Hand confidence |
|:------------------------:|:----------------:|:-----------------:|:-----------------:|:---------:|:---------------:|
| HOI-Swap (DINO encoder)  |       91.2       |       98.1        |       88.9        |   79.9    |       96.2      |
| HOI-Swap (CLIP encoder)  |       91.4       |       98.0        |       89.9        |   79.0    |       96.6      |

---

We would again like to thank all reviewers for their time and feedback, and we hope that our responses adequately address all concerns. Any further questions are highly welcomed.

---

> ### Comment · Area_Chair_3Gzg · 2024-08-08
> **Author-Reviewer Discussion Begins!**
>
> Dear authors,
>
> Thanks for your hard work and detailed responses.
>
> Dear reviewers,
>
> the authors have responded to your questions and comments, please read them and further provide feedback: Whether your concerns addressed? After reading all the reviews and responses, are there more questions about the clarity, contributions, results, etc?
>
> Thanks!
>
> Best, Your AC

---

> > ### Comment · Area_Chair_3Gzg · 2024-08-12
> > **Reminder: author-reviewer discussion will end soon!**
> >
> > Dear reviewers,
> >
> > The authors have responded to your questions and comments. Please propose your post-rebuttal comments.
> >
> > Best,
> > Your AC

---

### Decision · Program_Chairs · 2024-09-25

**Decision:**

Accept (poster)

**Comment:**

The paper studies the problem of swapping objects in hand-object interaction videos. The authors present a diffusion-based video editing framework. Comprehensive evaluations show that HOI-Swap outperforms existing methods, delivering high-quality video edits. Three reviewers think that their concerns are mostly addressed, and one reviewer further proposed that this paper should clarify its downstream scenarios further explain its capability boundaries, and raise the scores. After reading this paper, reviews, and responses, the AC agrees that this work made a good contribution.